# Systematic identification of cell size regulators in budding yeast

Ilya Soifer[†] & Naama Barkai[*]

## Abstract

Cell size is determined by a complex interplay between growth and division, involving multiple cellular pathways. To identify systematically processes affecting size control in G1 in budding yeast, we imaged and analyzed the cell cycle of millions of individual cells representing 591 mutants implicated in size control. Quantitative metric distinguished mutants affecting the mechanism of size control from the majority of mutants that have a perturbed size due to indirect effects modulating cell growth. Overall, we identified 17 negative and dozens positive size control regulators, with the negative regulators forming a small network centered on elements of mitotic exit network. Some elements of the translation machinery affected size control with a notable distinction between the deletions of parts of small and large ribosomal subunit: parts of small ribosomal subunit tended to regulate size control, while parts of the large subunit affected cell growth. Analysis of small cells revealed additional size control mechanism that functions in G2/M, complementing the primary size control in G1. Our study provides new insights about size control mechanisms in budding yeast.

**Keywords** cell growth; size control; START; yeast genetics
**Subject Categories** Genome-Scale & Integrative Biology; Cell Cycle
**Mol Syst Biol. (2014) 10: 761**

## Introduction

How cell size is determined is a fundamental question in cell biology. In most cell types, cell size is set by a feedback between the cell division cycle and mass growth (Jorgensen & Tyers, 2004; Turner *et al*, 2012). The specifics of this feedback differ between organisms. For instance, fission yeast delay mitosis until crossing some characteristic cell size (Nurse & Thuriaux, 1977). Mammalian leukocytes have to reach a certain specific growth rate, and epithelial cells need to grow to a certain size before committing to division (Dolznig *et al*, 2004; Tzur *et al*, 2009; Son *et al*, 2012). In budding yeast, cell size affects the duration of G1, with cells that are born small delaying the commitment to DNA replication and budding (START) (Johnston *et al*, 1977; Lord & Wheals, 1981; Di Talia *et al*, 2007).

It is convenient to think of the size control mechanism in budding yeast as a size-dependent checkpoint: a cell crosses START only when its size exceeds some threshold (Johnston *et al*, 1979). The real picture is, however, more complex. The compensation for size fluctuations is only partial, so that the size at START is not fixed but depends on cell size at birth and on its volume growth rate (Johnston *et al*, 1979; Ferrezuelo *et al*, 2012; Turner *et al*, 2012). Other models may fit the central observation that G1 phase is extended in small cells; for example, birth size could directly define G1 duration, the decision to cross START may depend on both the current size and the time elapsed from birth, or accumulation of a certain volume before undergoing START may be required (Aldea *et al*, 2007; Barberis *et al*, 2007; Amir, 2014).

The emerging molecular model of size control in budding yeast is centered on the SBF/MBF transcription complex, whose activity is inhibited by the transcription repressor Whi5 that binds SBF in late M/early G1 (Jorgensen *et al*, 2002; Costanzo *et al*, 2004; De Bruin *et al*, 2004). It is suggested that cell size is communicated by the G1 cyclin Cln3 (Cross, 1988; Futcher, 1996; Wijnen *et al*, 2002): Size-dependently accumulating Cln3 binds Cdc28 to inactivate Whi5, allowing START crossing and activating a stabilizing positive feedback (Skotheim *et al*, 2008; Charvin *et al*, 2010). Multiple mechanisms modulate Cln3 function: For instance, the RNA-binding protein Whi3 inhibits Cln3 translation (Garí *et al*, 2001), and Cln3 is retained in the ER by a chaperon Ydj1 through interaction with Whi7 (Vergés *et al*, 2007; Yahya *et al*, 2014). Finally, the dynamics of the START transition depends on additional pathways, the most studied of which works through Ccr4-Not complex and *BCK2* (Di Como *et al*, 1995; Manukyan *et al*, 2008).

It is not clear how size is measured and how the characteristic size is determined, but protein translation appears to play a central role (Johnston *et al*, 1977; Popolo *et al*, 1982; Moore, 1988). Both the rate of translation and the rate of ribosomal biogenesis affect cell size (Jorgensen & Tyers, 2004; Bernstein *et al*, 2007; Moretto *et al*, 2013). Translation initiation may be of a particular relevance: Nutrient starvation inhibits translation initiation through inhibition of TOR signaling which is thought to explain the small size of starved cells (Barbet *et al*, 1996), and mutations in translation initiation factors arrest yeast cells in G1 (Hanic-Joyce *et al*, 1987; Brenner *et al*, 1988; Anthony *et al*, 2001). Finally, the upstream ORF in *CLN3* mRNA limits its translation and could make its levels exceedingly

Department of Molecular Genetics, Weizmann Institute of Science, Rehovot, Israel
[*]Corresponding author. Tel: +972 8934 4429; E-mail: naama.barkai@weizmann.ac.il
[†]Present address: NRGENE LTD, Ness Ziona, Israel

sensitive to the overall rate of translation initiation (Polymenis & Schmidt, 1997).

Many important regulators of the size control were found using systematic screens for mutants that change the size distribution in cell populations (Jorgensen *et al*, 2002; Zhang *et al*, 2002; Ohya *et al*, 2005; Dungrawala *et al*, 2012). There are three limitations in screens of this type. First, size control in budding yeast occurs almost exclusively in the daughter cells constituting only approximately half of the cell population (Di Talia *et al*, 2009). Second, and more importantly, the characteristic size may be affected indirectly, not because of the effect of mutation on the size control mechanism. For instance, strains growing slowly may have a lower average size because the newborn cells are smaller. Similarly, strains with mutations extending the budded period produce large buds and therefore are larger. Finally, proliferation rate itself affects size control: Cells become larger as the proliferation rate increases. Thus, the vast majority of hits found in previous screens were genes that could not be directly linked to size control. These limitations may be overcome by quantitative time-lapse microscopy that follows cells throughout the cell cycle (Kang *et al*, 2014).

We describe a systematic screen designed to define processes affecting size control in budding yeast. We used high-throughput microscopy to follow millions of individual cells, defining their size at different cell cycle phases and the duration of these phases. A total of 521 candidate mutant strains were analyzed, chosen based on a high-throughput pre-screen and the existing literature. Our analysis neutralized confounding factors, such as differences in growth rate and initial cell size, allowing us to distinguish between mutants that affected size indirectly from those that affected the size control mechanism itself.

# Results

## Selection of candidate genes affecting size control

The majority of regulators of size control were found by searching for mutants that modulate cell size. Previous systematic screens measured size distribution in cell populations (Jorgensen *et al*, 2002; Zhang *et al*, 2002; Ohya *et al*, 2005; Dungrawala *et al*, 2012). We extended those "snapshot" screens to estimate not only the mean cell size, but also the relative duration of each cell cycle phase. To this end, we used a flow cytometer to estimate cell size (by the forward scatter) and to define the distribution of cells between different cell cycle phases (using an optimized DNA staining protocol; Fig 1A and B). Overall, we surveyed 96% of deletions of non-essential genes.

As expected, the majority of small mutants lacked components of the ribosomes ($P < 10^{-10}$) or were deleted of genes involved in ribosomal assembly and biogenesis ($P < 10^{-4}$). The mitochondrial ribosomes did not affect cell size or cell cycle in this screen unlike in previous screens (Jorgensen *et al*, 2002), likely reflecting the reduced respiration in our 96-well plate growth setup (Warringer & Blomberg, 2003). The largest cells were mostly perturbed in cell cycle progression (Supplementary Fig S1, Supplementary Dataset S1). We verified the reproducibility of our measurements by repeating the analysis for 750 small and 750 large strains (Fig 1C) and compared our results to previous screens (Fig 1D, Supplementary Fig S1F and Supplementary Dataset S1). 23 of the 26 strains

previously assigned the *whi* (small size) phenotype had average size below median ($P < 10^{-5}$) and one (*ygr064w*) did not grow well (Fig 1D and E). Overall, correlations between results of different screens were significant, but relatively low, stressing the difficulty of measuring cell size in high-throughput manner and the strong effect of environmental conditions on the average cell size.

To select candidates for size control regulators, we first examined the phenotype of known regulators. Deletion of *WHI5*, an inhibitor of START, reduced the cell size and decreased the frequency of G1 cells, suggesting shortening of this phase. Notably, this phenotype was distinct from that of small slow-growing strains which prolong (rather than shorten) G1 due to their small birth size. Similarly, deletion of *BCK2*, an activator of START, enlarged cells and showed a higher percentage of cells in G1, suggesting an extension of this phase (Fig 1A and B). Also here, this phenotype was different from that of mutants that overgrow during the S/G2/M phases, which are expected to have a shorter G1. Each of the 4,700 mutants analyzed was therefore characterized by its cell size and by the fraction of cells in G1. We selected strains with small size and relatively short G1 as candidates for being negative regulators and strains with a large size and relatively long G1 as candidates for being positive regulators (Fig 1F, Supplementary Text section 4). To overcome noise in size measurements, we used size estimations either from our pre-screen and its repeats or electronic volume measurement data from the screen by Jorgensen *et al* (2002). This way, we defined 255 putative negative and 264 putative positive regulators. We supplemented this list by strains involved in the ribosomal biogenesis and additional strains previously implicated in the regulation of START. Overall, a list of 591 candidate strains was assembled (Supplementary Dataset S2).

## Quantitative measurement of size control

To characterize size control in the candidate strains more directly, we developed a high-throughput video microscopy system that enabled following unperturbed growth and division of thousands of cells in parallel (Paran *et al*, 2007). Cells were grown on agarose pads and followed for 6 h at 3-min time resolution (~four cell divisions). To facilitate image analysis, we labeled the bud neck and the nucleus using the two fluorescence fusions Cdc10-GFP and Acs2-mCherry, respectively (Bean *et al*, 2006). Appearance of the bud neck marks the beginning of budding (synchronized with the beginning of S phase), while nuclear separation marks the metaphase to anaphase transition. Size regulation affects the period between birth and the Whi5 nuclear exit, while the subsequent time period is independent of size. However, since properties of size control are similar if measured at budding or, for example, via Whi5 localization (Di Talia *et al*, 2007), we decided to use bud emergence as a reporter to START, since its strong signal enabled automated detection and analysis. In a typical experiment, we followed 60 fields of view (up to 12 different strains) (Fig 2A) and used an image analysis software that we developed to automatically track individual cells, identify the cell cycle transitions and build cell lineages (Fig 2B and C, Supplementary Fig S2).

Size control is required to buffer fluctuation in size when cell volume grows exponentially, namely when the rate of volume increase is proportional to the cell volume. Whether budding yeast grow exponentially is debated (Di Talia *et al*, 2007; Goranov *et al*, 2009; Ferrezuelo *et al*, 2012; Hoose *et al*, 2012). To examine this in

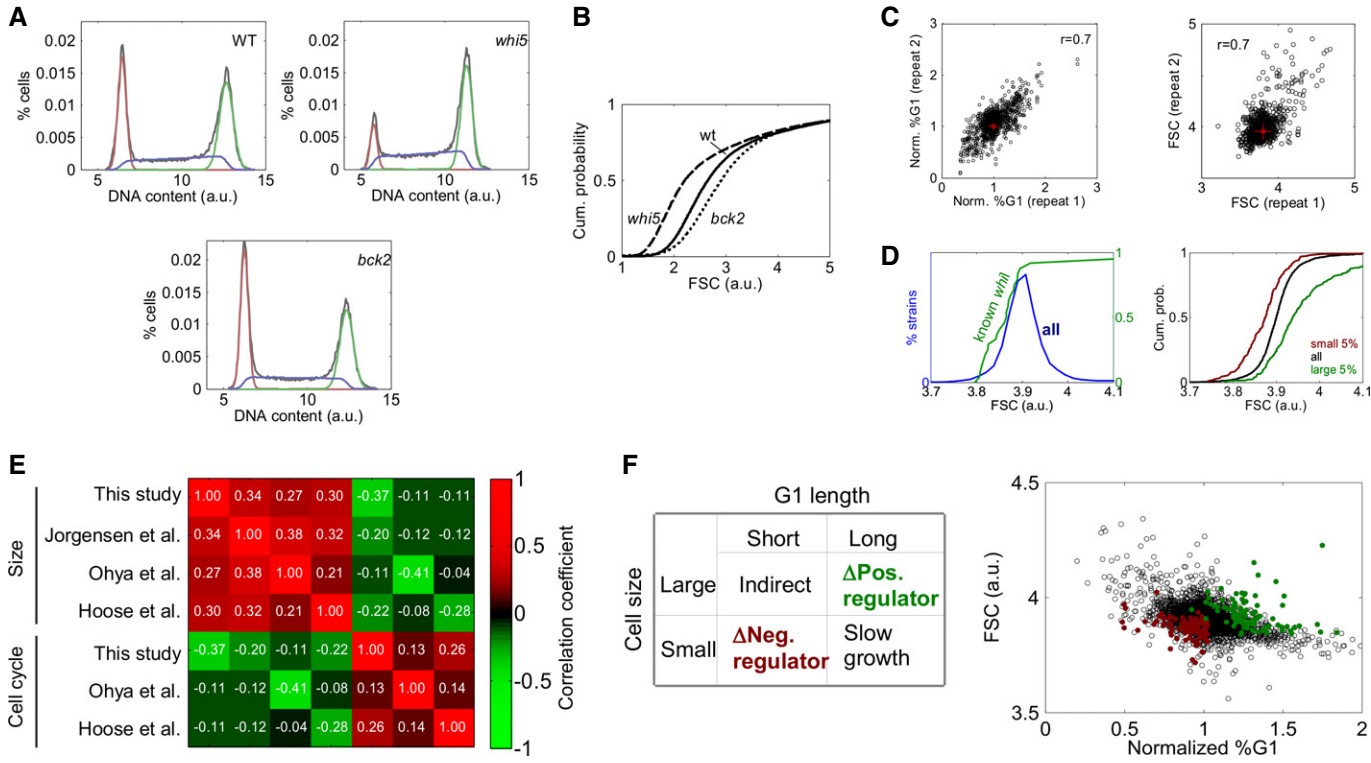

**Figure 1. Flow cytometry pre-screen.**

A   Distributions of DNA content in the wild-type (WT) and the two prototype deletions of positive (*bck2*) and negative regulator (*whi5*) of size control. Shown is the histogram of DNA content of logarithmic populations together with fitted distributions of cell cycle phases: G1 (red), S (blue) and G2 (green).

B   Cumulative distributions of forward scatter of WT, *whi5* and *bck2*.

C   Correlations between the measured percentages of G1 cells and of forward scatters between the two repeats of the screen.

D   Median FSC of all mutants and cumulative distribution of average forward scatters of mutants previously classified *whi* by Jorgensen *et al* (2002) and cumulative size distributions of the largest and the smallest 5% mutants previously found (left panel). Cumulative distributions of median FSC of all mutants, smallest 5% of mutants and the largest 5% of mutants from Jorgensen *et al* (2002) (right panel).

E   Pearson correlations between the median forward scatters/microscopic volume estimates/electronic volume estimates from this screen and previous screens (Jorgensen *et al*, 2002; Ohya *et al*, 2005; Hoose *et al*, 2012) and between percentages of G1 cells or percentage of unbudded cells measured in this screen and previous screens.

F   Classification of the candidate strains based on the cell size and the cell cycle phenotype. Small strains with short G1 (similar to *whi5*) were classified as deficient of negative regulators (red dots), and large strains with long G1 (similar to *bck2*) were classified as positive regulators (green dots). Note that the strains were selected based on the repeats and on the results of the pre-screen (see Supplementary Text section 4 for details).

our data, we tried to fit the increase in cell volume to a linear or to an exponential function. In support of the notion of exponential growth, when linear fit was attempted, the estimated growth rate was proportional to birth size, while no such correlation was observed when attempting an exponential fit (Supplementary Fig S3A and B).

Size control is often quantified by examining the (log) volume increase during G1 ($\Delta V \equiv \log V_s - \log V_b$) as a function of the (log) cell size at birth ($V_b$) (Sveiczer *et al*, 1996; Di Talia *et al*, 2007; Turner *et al*, 2012). In the absence of size control (and assuming exponential growth), this added (log) volume does not depend on birth size. Compensation for size fluctuations requires that more volume will be added to small-born cells, leading to negative correlation between the added volume and birth size. The extreme case of a perfect size control (checkpoint) predicts a slope of −1, as the final volume is independent of the initial volume. Consistent with previous observations, the added volume was negatively correlated with birth size in wild-type daughter cells. The slope of the negative

correlation was higher than −1, indicating a "weak" size control in which budding size is correlated with the size at birth (Lord & Wheals, 1981; Di Talia *et al*, 2007) (Fig 2D and E for diploids and haploids, Supplementary Table S1).

A complementary way for measuring size control is to examine how G1 duration depends on birth size. Also here, a negative correlation between G1 duration and cell size was observed, reflecting the prolonged G1 of small-born daughter cells (Fig 2F and G).

## Duration of G1 depends primarily on birth size and not on cell growth rate

Many of the candidate strains show a reduced growth rate. While growth rate affects cell size, we wanted to identify mutants that affect cell size in a way that could not be explained by their effect on the growth rate. This required some kind of normalization. In principle, different models of size control would make different predictions about how to perform this normalization. In the

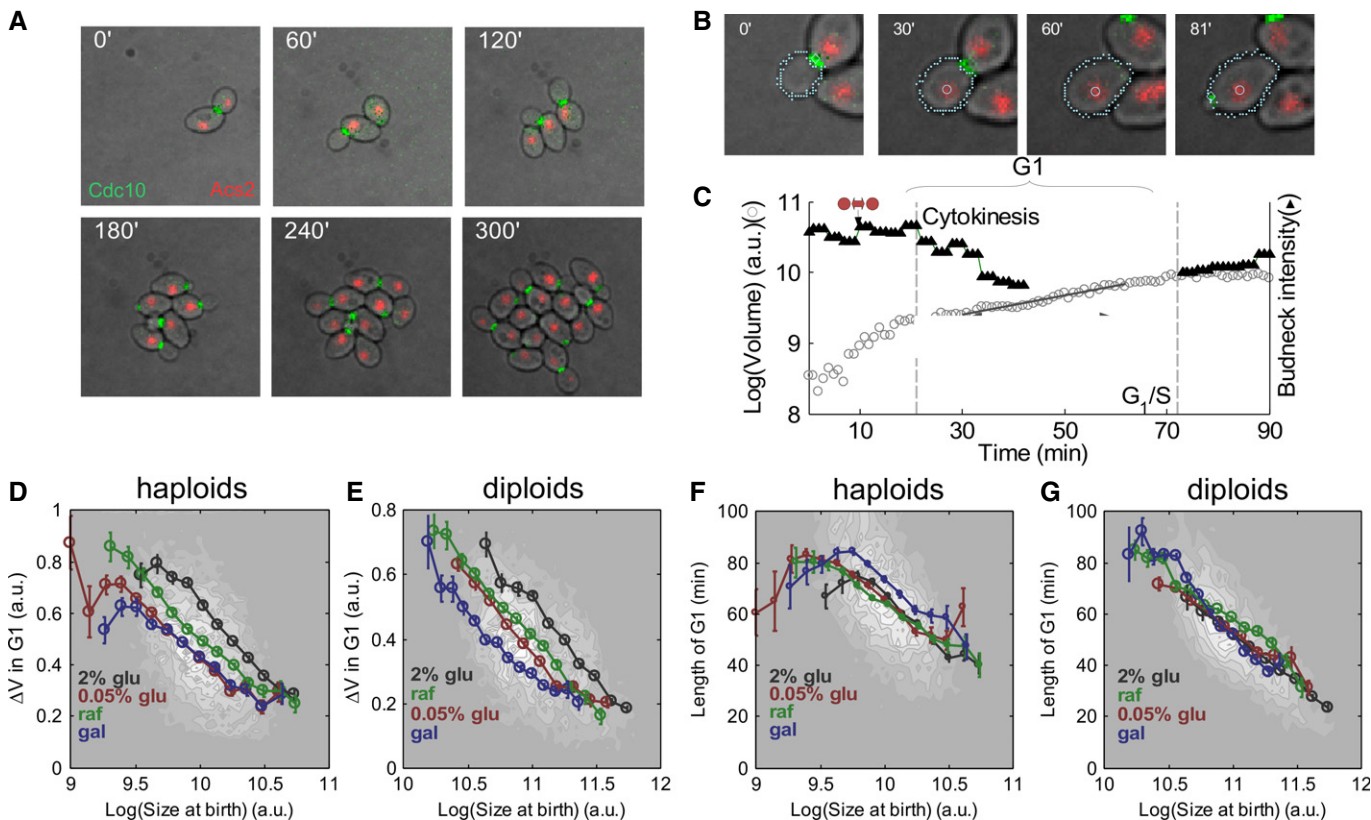

**Figure 2.** *In vivo* monitoring of division pattern in budding yeast reveals weak size control on glucose and at lower growth rates.

A Live imaging of multiple division cycles: composite image showing wild-type cells expressing Cdc10-GFP (green, bud neck) and Acs2-mCherry (red, nucleus) growing in our setup. We confirmed that in our setup the phototoxicity was minimal (Supplementary Fig S2A).

B Automated image analysis for tracking cells over time: composite image showing wild-type cells as in (A) with the contours found by the automated image analysis. Circle denotes the nucleus.

C Tracking cells allows for automatic determination of cytokinesis, START and the specific growth rate in G1. Shown is the volume as a function of time (circles) and the intensity of the bud neck (triangles) of a representative cell measured with a time resolution of 1 min. Gray lines denote cytokinesis and START (bud neck appearance), and red circle denotes time of nuclear separation. See Materials and Methods for details of determination of bud neck disappearance and appearance.

D, E Properties of the size control at different growth rates. log(size at birth) versus ΔV in G1 for haploid (D) and diploid (E) cells on glucose, low glucose (0.05%), raffinose and galactose. Black and white map shows two-dimensional histogram of all cells. Lines show data where cells from the same condition were binned into equally spaced bins along the x-axis. Cells on different media born at the same size bud at different sizes, consistent with different average specific growth rates. See Supplementary Fig S3B for direct comparison of the average budding size of cells born at the same size. See Supplementary Table S2 for statistical analysis.

F, G Duration of G1 has the same dependency on the size at birth in rich glucose, low glucose, raffinose and galactose: Log(size at birth) versus length of G1 for haploid cells (F) and diploid cells (G) on different media. Black and white map shows two-dimensional histogram of all cells. Lines show data where cells from the same condition were binned into equally spaced bins along the x-axis. Note that cells on different media born at the same size have almost indistinguishable duration of G1. See Supplementary Fig S3C for a direct comparison of the average duration of G1 of cells born at the same size. See Supplementary Table S2 for the details of statistical analysis.

checkpoint model, for example, if the threshold does not depend on growth rate, no normalization is necessary. Alternatively, if the threshold depends on growth rate, this dependency should be normalized for. Other models would make different predictions. We therefore decided to employ an empirical approach by comparing wild-type cells growing at different rates due to cell-to-cell variability, or differences in carbon source.

Cell size at budding increased with growth rate, as reported previously (Ferrezuelo *et al*, 2012; Supplementary Fig S3D). Consistently, cells born at a given size added more volume between birth and budding when provided with media supporting a higher proliferation rate (note the upward shift of the curves corresponding to different media in Fig 2D and E and Supplementary Table S2) (Johnston *et al*, 1979; Ferrezuelo *et al*, 2012).

In contrast, the duration of G1 was largely independent of growth rate, once birth size was controlled. Thus, cells that were born at the same size spent the same (average) time in G1 independently of their growth rate (Fig 2F and G, Supplementary Fig S3E, Supplementary Table S2). Consequently, slow-growing cells budded smaller, as in this same time they added less of a volume. This also caused the average G1 duration to increase with decreasing proliferation rate, as the average birth size decreased. We conclude that comparison of size control between strains growing at different rates is best done by quantifying the dependency of G1 duration on birth size, which is independent of the cell proliferation rate (at least for the doubling times of 86–124 min, which includes practically all mutant strains in our study, Supplementary Fig S3F).

 

## Microscopic screen for size control regulators

We next applied our microscopic setup to follow growth and division of the candidate mutant strains. The two fluorescent markers labeling the bud neck and the nucleus were introduced into the mutants using the SGA technology (Schuldiner *et al*, 2006; Tong & Boone, 2007). In general, strains that had a perturbed cell cycle maintained their phenotype following the SGA procedure. 70 strains that systematically did not retain the cell cycle phenotype or did not create viable SGA products were discarded from the analysis. (Supplementary Dataset S2)

In the first round, we followed at least 100 daughter cells from each strain and identified possible hits that perturb the size at budding. Suspected hits were repeated, with data taken for at least 300 daughter cells. Reproducibility between the two rounds was high (Fig 3A). Overall, we assayed 521 strains (Supplementary Dataset S3). The average size at budding was highly reproducible in 61 independent measurements of wild-type cells ($32.7 \pm 1$ fl, noise of $3\%$). Compared to this, many of the mutants had an altered budding size (Fig 3B). In particular, of the 21 previously defined *whi* mutants included in the screen, 19 had an average budding size that was 10–25% lower than wild-type (Supplementary Table S3). Average birth sizes reasonably correlated ($r = 0.45$) with birth sizes estimated indirectly from population data (Truong *et al*, 2013).

Over half of the strains that budded at a small size were depleted of elements of the translation machinery. Inhibitors of START and mediators of glucose signaling were also included in this group (Fig 3C). Genes whose deletions increased cell size belonged to diverse functional groups (Fig 3D). Importantly, as described above, large or small budding size is not, by itself, an indicator of an altered size control.

## Classification of size-perturbing mutants

To define mutants affecting the size control mechanism, we examined how G1 duration depends on birth size and how the increase in volume during G1 depends on birth size (Fig 4). We classified all mutants into nine categories with respect to extended/shortened/normal G1 duration and increased/decreased/normal $\Delta V$ in G1 (Fig 4). This classification was done by dividing the cells into evenly spaced bins according to their birth size, calculating the average G1 duration in each bin and the average volume increase. These values were compared to the corresponding wild-type values, and $P$-value for the difference was calculated (see Materials and Methods and Supplementary Fig S4). Thus, we asked whether cells born at a given birth size spent longer/shorter time in G1 (or grow more/less) compared to wild-type cells born at the same size. About two-fifths (197) of the strains showed statistically significant ($P$-value < 0.001)

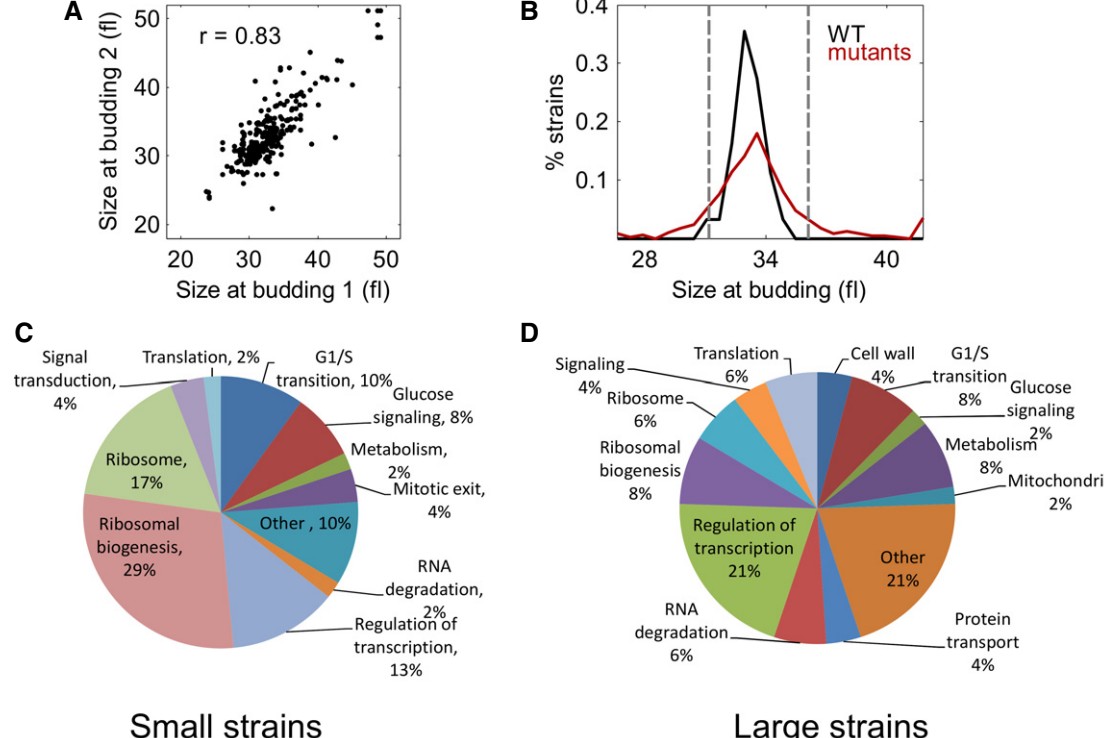

Small strains                 Large strains

**Figure 3.  Summary of the microscopic screen.**

A    Reproducibility of measurements of size at budding. Median budding size of the same strain between the two repeats of the measurement.

B    Many mutant strains have a perturbed size at budding. Histogram of cell sizes at budding of mutant strains versus the cell size at budding of the 60 repeats of the wild-type strain.

C, D    Small-budding strains are mostly deficient in the elements of translation machinery, while large-budding strains belong to diverse functional groups. Pie charts showing the function of deletions in cells that bud small (C) and large (D).

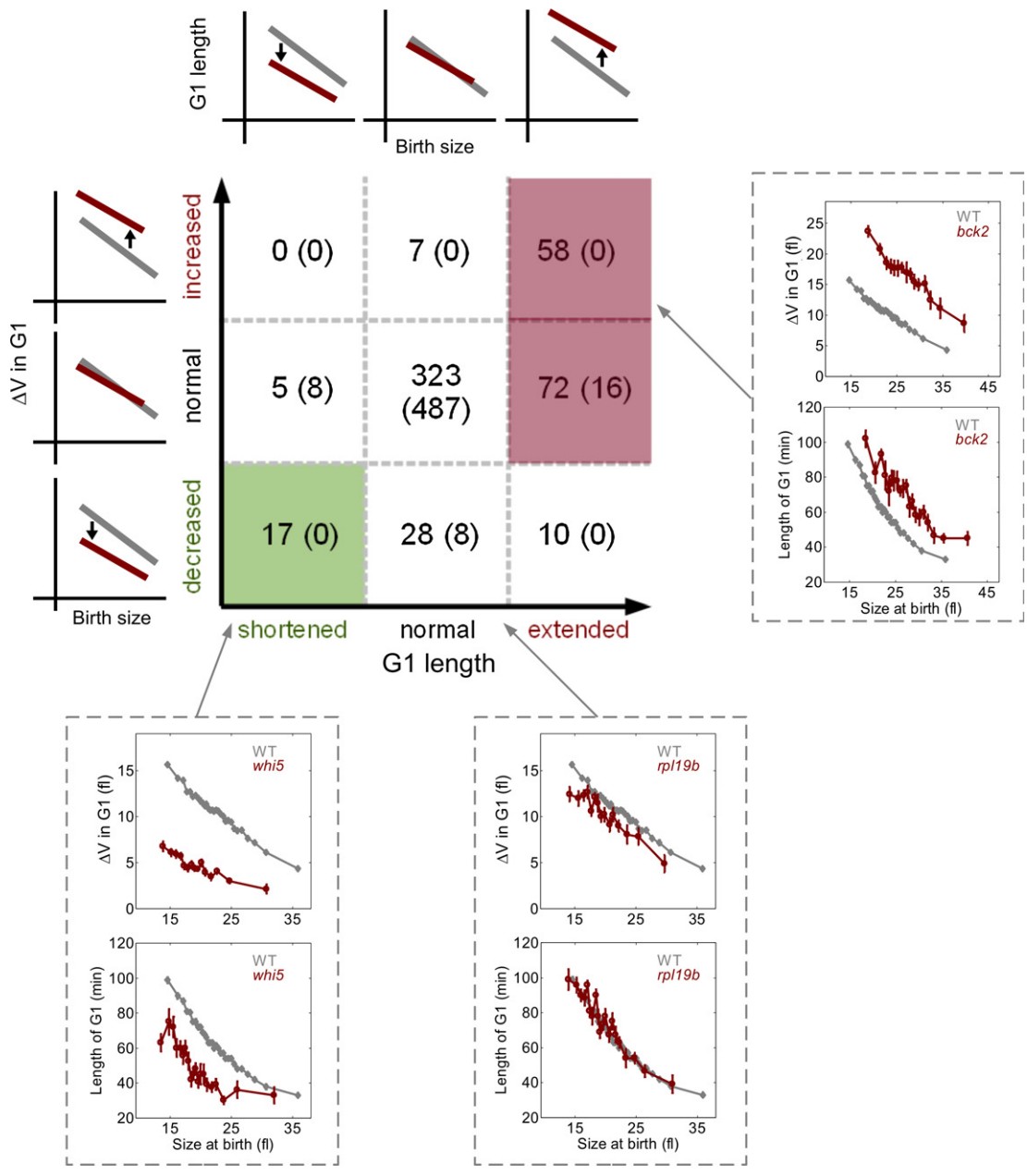

**Figure 4.  Classification of the mutant strains.**

Classification of the mutant strains according to the dependency of length of G1 and volume increase (ΔV) in G1 on the birth size. Strains fell into one of the nine categories with shorter and longer G1 and decreased and increased ΔV in G1. See Materials and Methods for the details of statistical analysis. Number of strains falling into each category with the estimated number of false positives is indicated. Insets show example strains in each category. Note that mutants having shorter G1 but normal volume increase were not classified as negative regulators, since they are all expected to be false positives.

difference from the wild-type (Supplementary Dataset S3). Based on the wild-type repeats, we estimate that at most 32 were false positives.

Most small strains (32/52) did not alter G1 duration (given their birth size), but their small size was explained by a reduced increase of volume during G1 (e.g., *rpl19b*). In other strains, growth in G1 was not reduced, but the bud grew less than expected in the budded phase (either due to shortening of the budded phase, e.g., *swe1*, or due to slower growth, e.g., *tom1* or *rpp1b*). The decreased bud

growth generated small newborn cells that budded at a smaller than normal size. Among nineteen previously identified *whi* strains, twelve belonged to this category, suggesting that their small size results from a slower growth rate and not from a direct perturbation of the size control mechanism. In contrast, the majority of the largest strains (34 of 50) had an extended G1 phase, suggesting that G1 delay is the primary cause for their larger size. The other large strains were born large but did modify their respective (normalized) G1 duration (Supplementary Fig S5).

Seventeen deletion strains shortened relative length of G1 and were therefore classified as negative regulators of START. 130 strains extended G1 relative to birth size and were classified as positive regulators of START. Note that since we measured G1 length and not the execution of START, some of the mutants could affect budding and not START. Most of the identified regulators, however, do not belong to functional categories that seem likely to decouple those two processes.

Some strains could not be assigned a category unambiguously. For example, ten strains had a smaller increase in volume during G1 relative to their birth size, but a significantly extended G1 (e.g., *sfp1*). All of these strains were characterized by a very slow growth rate. While those strains naturally fall into our definition of positive regulators, we were careful in making this assessment, because of their very slow growth rate, which falls out of the growth rate intervals for which we observed an independency of G1 duration on growth rate.

### Negative regulators connect G1 duration to the mitotic exit/polarity establishment network

The smallest of the negative START regulators were the known effectors *whi5* and *ydr417c* (Table 1, Fig 5, Supplementary Fig S6). Since *ydr417c* is a partial deletion of ribosomal protein Rpl12b, we were surprised that the full deletion of *RPL12B* did not shorten G1 (Supplementary Dataset S3). Notably, *ydr417c* was previously shown to significantly increase chronological lifespan, a phenotype connected to cell cycle control (Fabrizio *et al*, 2010).

To examine for common properties of deletions assigned to this group, we analyzed their interaction network using the physical and genetic interactions described in the BioGRID database (Stark *et al*, 2011). In this analysis, we considered *ygr151c*, a partial deletion of *BUD1/RSR1*, as representing the *rsr1* (which was absent from our screen), as it showed the random budding pattern characteristic of *RSR1* deletion (Bender & Pringle, 1989). Two connected components emerged from this analysis (Fig 5D). The first contained known nutrient-dependent regulators of START: the glucose signal receptors *GPA2*, *GPR1* and the RNA-binding protein *WHI3* that plays multiple roles in starvation (Garí *et al*, 2001; Alberghina *et al*, 2004; Colomina *et al*, 2009).

The second, larger, connected component contained seven genes: three previously identified as negative regulators of START, *CDH1*, *SIC1*, *WHI5*, three novel regulators, *LTE1*, *YGR151C/RSR1* and *STE20*, and Whi2, a START regulator previously implicated in stationary phase only (Saul & Sudbery, 1985) (Fig 5D). Notably, this interacting component was associated with the mitotic exit network, suggesting that mitotic exit is involved in setting the duration of the ensuing G1.

Negative regulators that do not belong to the connected components included a known effector, *KAP122*, and several novel ones: *PAT1*, *MED1*, *YLR112W*, *SEL1*, *YDR417C* and *LRE1*.

### Positive regulators of START

The class of positive regulators, whose deletions prolonged G1, included all seven known effectors of the G1/S transition present in

**Table 1.   List of identified negative regulators.**

| Systematic name | Name | Known function | Known function | Size at budding (pxl$^3$) | Relative G1 (min) | Relative G1 percentage |
|---|---|---|---|---|---|---|
| YOR082C[a] | | G1/S transition | Overlaps *whi5* | 1213 ± 319 | −31 | 0.5 |
| YOR083W | WHI5 | G1/S transition | Repressor of late G1 transcription | 1273 ± 278 | −26 | 0.5 |
| YDR417C | | Ribosome | Partial deletion of ribosomal subunit | 1298 ± 462 | −14 | 0.95 |
| YNL197C | WHI3 | G1/S transition | Repressor of translation of G1 cyclins | 1433 ± 529 | −17 | 0.98 |
| YGL016W | KAP122 | Other | Karyopherin, nuclear transport | 1476 ± 270 | −9 | 0.94 |
| YDL035C | GPR1 | Glucose signaling | Glucose receptor, cAMP signaling | 1531 ± 248 | −5.5 | 0.98 |
| YER020W | GPA2 | Glucose signaling | cAMP signaling | 1548 ± 358 | −6 | 1.15 |
| YGL003C | CDH1 | Mitotic exit | Mitotic cyclin degradation | 1569 ± 835 | −12 | 0.76 |
| YLR079W | SIC1 | G1/S transition; Mitotic exit | Inhibition of mitotic and late-G1 cyclins | 1569 ± 786 | −12 | 0.2 |
| YCR077C | PAT1 | RNA degradation | Decapping and deadenylation of mRNA | 1569 ± 277 | −17 | 1.26 |
| YML013W | SEL1 | Other | Protein degradation | 1617 ± 411 | −8 | 0.71 |
| YHL007C | STE20 | Mitotic exit | Kinase involved in cell growth and mitotic exit | 1618 ± 331 | −10 | 0.54 |
| YLR112W | YLR112W | Other | Unknown | 1649 ± 394 | −10 | 1.15 |
| YOR043W | WHI2 | G1/S transition | Stress response, growth during stationary phase | 1663 ± 322 | −3.5 | 0.65 |
| YAL024C | LTE1 | Mitotic exit | Part of MEN network that regulates mitotic exit | 1668 ± 371 | −9 | 0.5 |
| YGR151C | YGR151C | Cell polarity; Mitotic exit | Overlaps *RSR1*, Involved in mitotic exit | 1680 ± 460 | −10 | 0.85 |
| YCL051W | LRE1 | Other | Cell wall maintenance | 1750 ± 391 | −10 | 0.99 |
| IS003 | WT | | | 1744 ± 356 | 0 | 1 |

[a]YOR082C is a dubious ORF that paritally overlaps *WHI5*. Since its deletion has a phenotype very similar to the deletion of *WHI5* we discarded it from the further analyses.

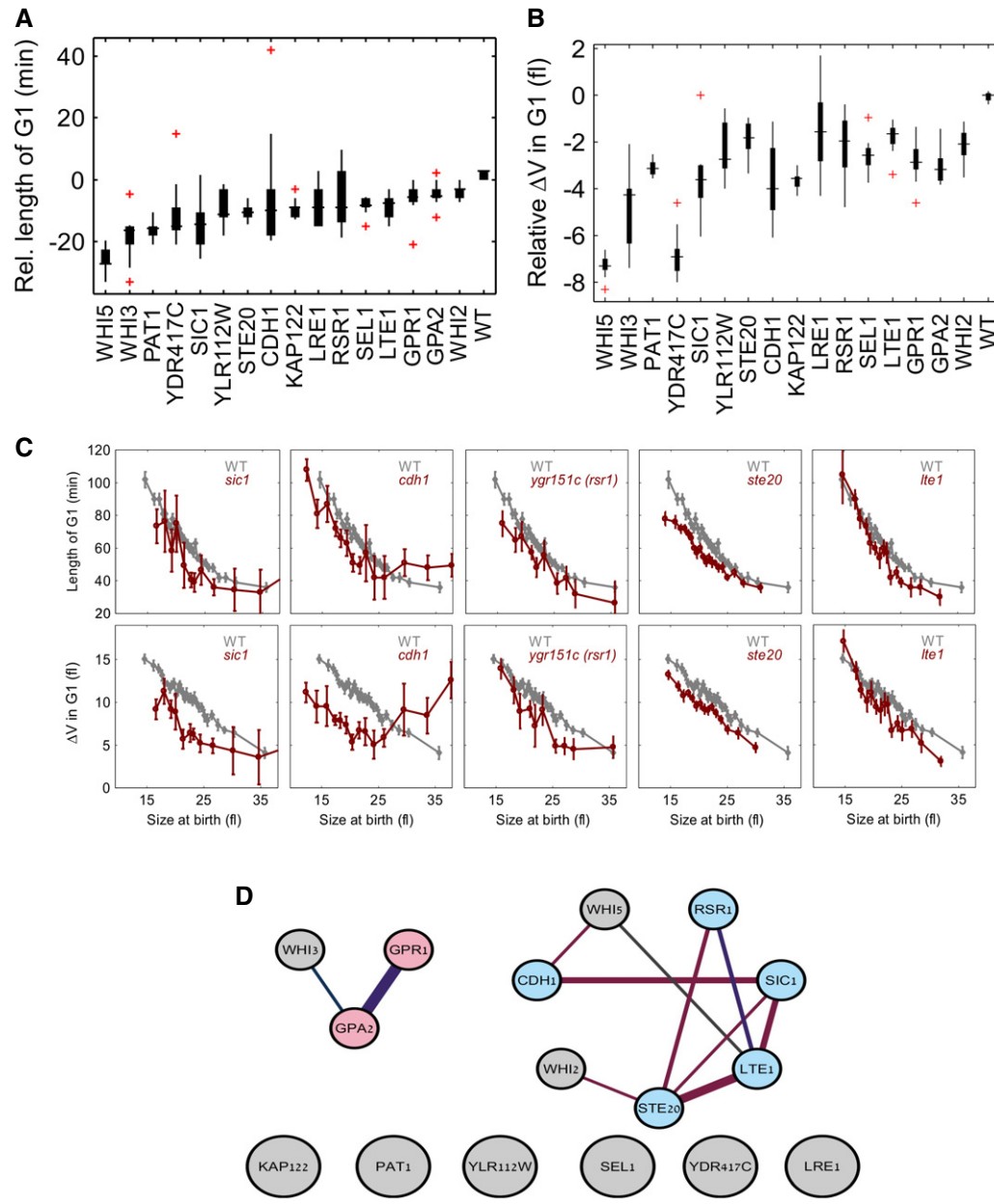

**Figure 5.   Negative regulators of the size control form a network of genetic interactions and belong to cell polarity and mitotic exit network.**

A   Box plot showing relative length of G1 in each size bin (median length of G1 for the strain − median length of G1 of wild-type cells born at the same size) in the mutants belonging to the negative regulator category compared to the wild-type. *yor082c* (ORF overlapping *WHI5*). Red plus markers denote outlying bins.

B   Relative ΔV in G1 (median ΔV in G1 for the strain − median ΔV in G1 of wild-type cells born at the same size) in the mutants belonging to the negative regulator category.

C   Dependency of the length of G1 and ΔV in G1 on the size at birth in the mutants that belong to the negative regulator category (representative mutants, average across size bin), see Supplementary Fig S6 for all mutants. At least 500 cells are measured for each mutant.

D   Genetic and physical interactions between negative regulators of START. Red line: genetic interaction, violet line: physical interaction. Blue nodes: mitotic exit/polar growth genes, pink nodes: glucose signaling genes.

our screen (*CLN2, CLN3, SWI4, BCK2, MBP1* and *RME1*). Many other positive regulators were linked to translation or to ribosomal biogenesis (21 strains). In addition, five other regulators were linked to various aspects of mitochondrial translation. Overall, translation emerged as the main process that stimulates START, thereby effecting G1 duration.

Chromatin remodelers were also found to affect START positively. This group included deletions of histone deacetylases *RXT3* and *SNT1* and deletions of histone ubiquitinases *RAD6* and *LGE1* (deletion of the *BRE1* ubiquitin ligase prolonged G1 but did not reach the required statistical significance). Genes involved in the cell wall integrity checkpoint, including *ROM2, SLG1, SSK1* and

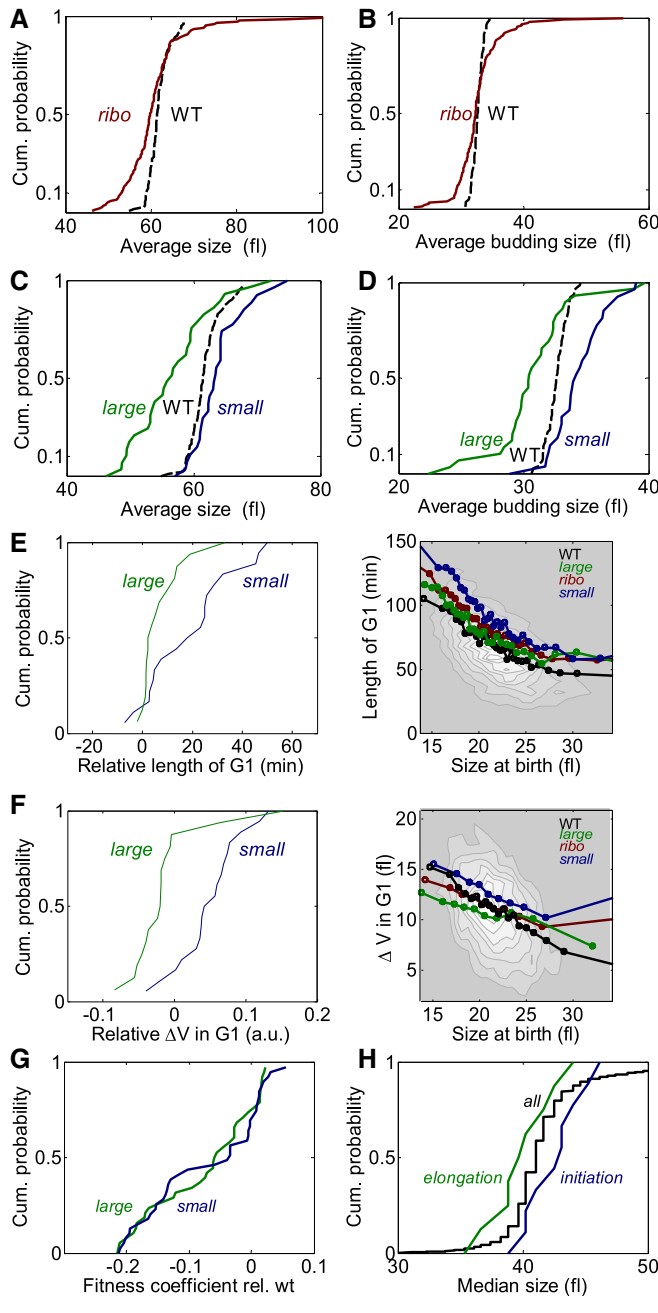

**Figure 6.  Protein synthesis has a positive effect on START.**

A, B    Deletions of ribosomal genes have a decreased average cell size (A) and a diverse average budding size (B). Cumulative distribution of average cell size (A) and average cell size at budding (B) for deletions of ribosomal and ribosomal biogenesis genes and the wild-type is shown.

C, D    Distinct size phenotypes of the deletions of large and small subunits. Cumulative distribution of average cell size (C) and cell size at budding (D) for deletions of small (blue) and large (green) ribosomal subunits compared to the wild-type is shown.

E, F    Different effect on the size control of deletions of the small and large ribosomal subunit. Left: Cumulative distributions of relative lengths of G1 phase (E) and relative $\Delta V_{G1}$ (F) of deletions of parts of the small and large subunit of the ribosome are shown. Right: Plots as in Fig 3E–G. Data for twelve deletions of factors in the small ribosomal subunit and nine factors in the large ribosomal subunit that have a significantly decreased growth rate (doubling time > 110 min) are plotted binned together or separately. Note that deletions of the large subunit do not affect the dependency of G1 on the birth size, while the deletions of parts of the small subunit extend G1. The surface plot shows all the data binned together. See also Supplementary Fig S7.

G    Similar effect of deletions of parts of the large and small ribosomal subunit on the cell growth rate. Cumulative distribution of growth rates of strains deleted in parts of the small and large subunit of the ribosome is shown.

H    Different effect of deletions of elongation and initiation factors on the characteristic cell size. Cumulative distribution of median cell sizes of deletions of the initiation and elongation factors is shown. Data from Jorgensen *et al* (2002).

slow-growing mutants (Fig 6A). Surprisingly, the effect on daughter cell budding size was less consistent: While some mutations decreased budding size, others led to its increase (Fig 6B).

It was previously shown that perturbing the small or the large ribosomal subunits results in distinct effects on cell size, cell cycle progression and bud morphology (Jorgensen *et al*, 2004; Hoose *et al*, 2012; Moretto *et al*, 2013; Thapa *et al*, 2013). We observed that deletions in the large subunit significantly decreased cell size, while deletions of parts of the small subunit had a small effect on the average size and tended to increase cell size at budding (Fig 6C and D; Supplementary Table S4). These effects were consistent when comparing our data to population data from other screens (Supplementary Fig S7A–C). To interpret these results, we examined the effects of deletions on G1 duration. Deletions in the large subunits had a very small effect on G1 duration, once birth size was controlled for (Fig 6E), but increased their volume less than expected from their birth size consistent with a slow growth rate (Fig 6F). In contrast, deletions of parts of the small subunit significantly increased the (birth size normalized) G1 duration, leading to daughter cells budding at larger size (Fig 6E and F). These observations were robust between repeated measurements (Supplementary Fig S7D) and statistically supported (Supplementary Table S4). We conclude that the small (but not large) ribosomal subunits act as positive regulators of START. Note that proliferation rate was similarly affected by deletions in the small and large subunit, as measured by a sensitive competition assay (Fig 6G) or lengths of mother cells in the microscopic screen (Supplementary Fig S7E). The average length of G1 also increased in both cases, since cells were born at a size that was smaller than that of the wild-type and thus budded after a longer time.

Both ribosomal subunits participate in translation elongation. However, translation initiation requires binding of the small subunit

*RRD2,* were also assigned to this group, suggesting that their deletion is sensed as cell wall damage thereby prolonging the G1 phase.

**Effect of translation and ribosomal biogenesis on START**

The role of translation in cell size regulation is disputed. Early studies suggested that translation capacity promotes START as inhibiting translation extends G1 and increases budding size (Hartwell & Unger, 1977; Popolo *et al*, 1982; Moore, 1988). It was later found, however, that strains deficient in ribosomal biogenesis are smaller than wild-type (Jorgensen *et al*, 2002, 2004; Moretto *et al*, 2013).

In our data, deletions of ribosomal genes or of ribosomal biogenesis genes substantially decreased the average cell size, especially in

first, followed by binding of the large subunit only upon transition to elongation. The distinct phenotypes of the large and small ribosomal subunit suggest that although growth rate depends on actual translation, the START transition is sensitive to the rate of translation initiation. Indeed, translation of all cyclins is highly sensitive to the rate of translation initiation (Barbet *et al*, 1996; Polymenis & Schmidt, 1997; Philpott *et al*, 1998). Consistent with this hypothesis, in a previous screen (Jorgensen *et al*, 2002), deletions of elongation factors decreased the average cell size (similar to deletions affecting the large ribosomal subunit), while deletions of initiation factors increased the size (similar to deletions affecting the small subunit) (Fig 6H).

**A size-regulating mechanism in the budded phase**

The overall doubling time of the mutants showing rapid progression through G1 was not reduced (Table 1), suggesting that other cell cycle phases are extended to compensate for the shortened G1. Extension of the budded phase can be explained in two ways. First, the overall duration of the cell cycle may be controlled, for example, by some process which is initiated at cell birth and has to be completed before cell divides. Alternatively, an additional size-dependent regulation may exist which is exposed when cells bud at a smaller size. In both cases, when G1 is shortened, other phases of the cell cycle are expected to be extended.

To distinguish between these two options, we examined whether the duration of G1 correlates with the duration of the budded phase. No correlation was observed in wild-type cells ($r = 0.026$), ruling out the hypothesis of a constant cell cycle time (Fig 7A). On the other hand, the duration of the budded phase was negatively correlated with the size at budding, suggesting that size control, albeit weak, is acting also in the budded phase ($r = -0.24$, slope = $-0.01 \pm 0.0014$ min/pxl$^3$, red line on Fig 7B). It was suggested that the primary size control in G1 is less effective in large cells compared to smaller ones (Di Talia *et al*, 2007). Focusing on mutants that bud at a small size, we observed that size control in the budded phase became significantly stronger (Fig 7C, $r = -0.35$, slope = $-0.038 \pm 0.004$ min/pxl$^3$ in *whi3* and $-0.025 \pm 0.003$ min/pxl$^3$ in *whi5*). Binned together, the data for the small mutants were very similar to the wild-type data in the region of the overlap (relatively large cells), but had a significantly higher slope for smaller cells. This effect was noticeable both when looking in daughter cells (Fig 7C, green line, $r = -0.42$, average slope = $-0.025$ min/pxl$^3$) and when examining mother cells (Supplementary Fig S8). Therefore, our data point to a size control mechanism that functions in the budded phase, which is weak in large cells, but becomes stronger when cells exit G1 as a small size.

# Discussion

**New classification of size control mutants**

Cell size is a complex phenotype that is controlled by a myriad of pathways and a complex interplay between growth and division. Despite decades of work, the molecular basis of size control is poorly understood. One of the challenges in the study of size control is the difficulty in distinguishing between mutations that directly

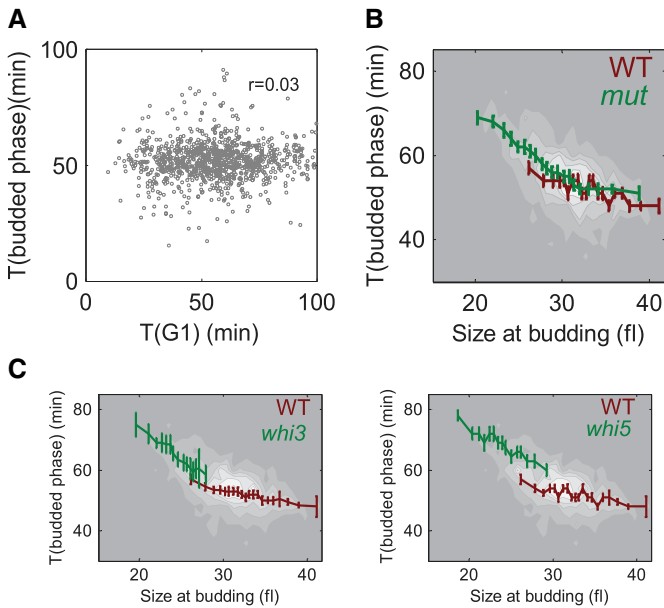

**Figure 7.  Backup size control in the budded phase of small mutants.**

A    Duration of the budded phase is independent of the duration of G1. Plotted is the duration of G1 versus the duration of the budded phase in wild-type daughter cells.

B, C  Small mutants have a stronger dependency of the length of budded phase on the size at budding than the wild-type. Data for the wild-type strain (red line) and small mutants (green line) were binned according to the size at budding, and the average duration of the budded phase was plotted. Small mutants that have cell cycles of a similar length to that of the wild-type were considered. The surface plot shows all the data binned together. In (C), only data for *whi3* and *whi5* are displayed.

affect the size control mechanism and mutations that affect size by changing the rate of volume growth. An additional complexity arises from the fact that the size control mechanism corrects size fluctuations only partially. Therefore, given that a gene is affecting the characteristic size in a cell population, it is difficult to associate its function with the size control mechanism directly. Single cell data is of immense value, as it allows focusing on the cells during the phase when size control is acting.

We performed a systematic screen for mutants affecting size control in the budding yeast. Building on previous screens and selecting additional candidate genes by high-throughput flow cytometry-based pre-screen, we measured size control in over 500 yeast deletion strains using live cell imaging, following cells for 6 h of unperturbed growth. A large proportion (about 40%) of strains that we examined had a significantly larger or smaller size compared to wild-type cells.

Examining size control in wild-type cells growing at different rates suggested a way to distinguish between the direct and the indirect (e.g., acting through growth rate) effects on size regulation. When considering cells born at the same size (but growing at different rates), the increase in volume during G1 was strongly dependent on the proliferation rate. In contrast, the duration of G1 was largely defined by the initial cell size, independently of the cell growth rate, at least for growth rate interval of 86–124 min which we have checked and where the vast majority of wild-type cells reside. We therefore considered the size-dependent regulation of G1 length as

    

the primary mode by which cells guard against size fluctuations, and examined for mutants that alter this dependency.

Using this measure, we found that the majority of the strains that are currently classified as small (*whi*) do not directly affect size control, showing the same wild-type dependency between birth size and G1 duration. Their small size results from slower growth rate and/or smaller birth size. As small birth size is an indicator of impeded cell cycle progression, these strains could be used to understand the mechanism of cell cycle progression (Truong *et al*, 2013).

## Role of mitotic exit network in the regulation of START

Our data implicated the mitotic exit network as an important point in size regulation. This suggests that START transition is not determined solely by the instantaneous state of the cell (e.g., size, protein synthesis capacity) but depends also on the previous cell cycle, prior to the completion of division and cell separation. The mechanism by which the mitotic exit and polarity establishment networks affect the START transition is still obscure. We note, however, that the involvement of identified regulators in START is supported by several previous studies: *CDH1* deletion leads to early budding at a smaller size (Jorgensen *et al*, 2002; Wäsch & Cross, 2002), and genetic interactions between *LTE1* and *WHI5* as well as between *CDH1* and *WHI5* were reported (Ye *et al*, 2005). Further, the mitotic exit network and the polarity establishment network are connected: Double deletion of *STE20* and *LTE1* is unable to exit mitosis (Höfken *et al*, 2002; Höfken & Schiebel, 2004), and Lte1 interacts physically with Rsr1 (Lai *et al*, 1993). Finally, polarity establishment network interacts tightly with START network as shown by various genetic interactions between *STE20* and *CLN2* and *CLN3* (Fiedler *et al*, 2009).

One possible model for the role of the mitotic exit in the START transition may involve the release of the Cdc14 phosphatase from the nucleolus as Whi5, the key inhibitor of START, is dephosphorylated by Cdc14 in late mitosis (Taberner *et al*, 2009). It is possible that impairing the mitotic exit network decreases the pulse of Cdc14 activity, thereby decreasing the amount of Whi5 that enters the nucleus at the end of cytokinesis and shortening the time until START.

## Role of protein synthesis in the regulation of START

Protein synthesis had long been implicated as a positive regulator of START (Popolo *et al*, 1982; Moore, 1988). Inhibition of protein synthesis delays START, causing cells to bud at a larger size. Although the details of how protein synthesis promotes START are not completely clear, at least partly it acts through Cln3. An upstream ORF in the *CLN3* mRNA inhibits its translation (Polymenis & Schmidt, 1997). Due to this upstream ORF, the translation of Cln3 is affected disproportionally relative to other proteins when the overall protein synthesis capacity is reduced.

As had been pointed out, this model is not without certain difficulties (Jorgensen & Tyers, 2004; Turner *et al*, 2012). If the overall translation rate stimulates START, one would expect that in poor growth conditions or when ribosomal content is decreased, cells would also delay START and increase their size. In general, however, the opposite is observed: Poor nutrient conditions (Johnston *et al*, 1979) or deletions affecting the ribosome

(Jorgensen *et al*, 2002, 2004; Yu *et al*, 2006) decreased the average cell size. This led to the suggestion that while translation itself is a positive regulator of START, the rate of the ribosomal biogenesis has a negative role in START (Jorgensen & Tyers, 2004). In particular, since deletions of proteins involved in the assembly of the large ribosomal subunit (structural proteins or biogenesis factors) decrease cell size to a larger extent than factors of the small ribosomal subunit, it was proposed that START depends on the flux through the pathway producing the large subunits (Dez & Tollervey, 2004; Moretto *et al*, 2013).

Our results suggest a unified explanation for those findings. Upon screening of approximately one-third of nonessential ribosomal proteins, we observed that deleting components of the large ribosomal subunit or of genes involved in ribosomal biogenesis does not affect the actual size control. Rather, cells become smaller simply because they grow slower. In contrast, parts of the small ribosomal subunit behave as positive regulators of START, extending G1 duration more than expected given their birth size, and consequently budding at a size comparable or larger than wild-type cells born at a small size. This likely reflects their distinct role in translation initiation, not shared by the large ribosomal subunit. Our results therefore suggest that translation initiation is a positive regulator of START, hence its inhibition, as observed in the initial experiments, prolongs G1 and could lead to a larger budding size. In contrast, translation elongation affects predominantly the cell growth rate and therefore decreases cell size, as observed upon deletion of ribosomal components.

We note that this role of translation initiation in promoting the START transition is supported by multiple studies: Deletion of eIF4 (*CDC33*) and eIF3 (*CDC63*) prolongs G1 and increases cell size (Hanic-Joyce *et al*, 1987; Brenner *et al*, 1988; Polymenis & Schmidt, 1997), and many strains depleted of translation initiation factors have an increased size (Fig 6H).

## Backup mode of size control

Mutants with a shortened G1 did not show an overall decrease in cell cycle time. This suggested that the other (budded) phases of the cell cycle are prolonged. Extension of the budded phase provided some compensation for difference in budding size, preventing birth of very small cells. By examining those small cells, we revealed an additional size-regulatory mechanism. In fact, for those small cells, the strength of this backup size control was approaching the primary size control that functions during G1.

Previous evidence suggested that G2/M morphogenesis checkpoint can also act as a cryptic size control, activated, for example, when the bud is not large enough (Rupes, 2002; Harvey & Kellogg, 2003; Anastasia *et al*, 2012; King *et al*, 2013). We do not know whether the phenomenon we observe is related to morphogenesis checkpoint. Our results are reminiscent of the cryptic size control point identified in fission yeast in small *wee1* mutants (Fantes & Nurse, 1978). Note that in the fission yeast, both the primary and the cryptic size control comply with the checkpoint paradigm (Sveiczer *et al*, 1996), while in budding yeast, both size controls compensate only partially for size fluctuations.

The strengthening of size control in small cells questions the classical distinction between "timers" and "sizers". Timers are phases of the cell cycle that do not depend on cell size, while sizers

are phases that are size dependent. Our results suggest that this distinction is arbitrary. It seems that all phases of the cell cycle could be timers or sizers depending on cell size. Perhaps when cell size is small, some cellular components are becoming limiting for cell cycle progression, making the length of this phase size dependent. In large cells, the same phase becomes a timer since these components are no longer limiting. In this model, cell size affects the rate of cell cycle progression, instead of being a requirement for transitions between phases, similar to models proposed mathematically (Chen *et al*, 2000; Pfeuty & Kaneko, 2007; Charvin *et al*, 2009). As previously argued, this alternative mode of size control is sufficient to ensure size homeostasis under conditions of exponential growth (Tyson & Hannsgen, 1985; Csikasz-Nagy *et al*, 2006). In the Supplementary Text section 5, we briefly analyze this model of size control and show that it ensures size homeostasis.

# Materials and Methods

### Strains

The wild-type haploid strain is alpha-type magic marker strain created from Y8205 (Tong & Boone, 2007) by fusing a C-terminal eGFP tag to *CDC10* and a C-terminal mCherry tag to *ACS2*. The size distribution and the durations of the cell cycle phases were indistinguishable in this strain from BY4741. Yeast deletion collection was obtained from EUROSCARF. The deletion strains containing Cdc10-GFP and Acs2-mCherry were obtained by SGA methodology as described in Schuldiner *et al* (2006).

### Flow cytometry screen

Preparation of cells for the flow cytometry was performed using robotic assay as described in Koren *et al* (2010). The populations were measured using LSRII flow cytometer with HTS attachment (BD Biosciences). SYBR green, FSC and SSC parameters were acquired; at most 30,000 events were acquired per well. Wells with less than 5,000 events were discarded from the analysis and re-run. Events with the fluorescence area below 5,000 or above $2^{18}$–5,000 and with top 1% and low 1% width of fluorescence peak were discarded. The remaining data were binned into 100 bins, smoothed using `csapi` MATLAB function; the lowest bin containing at least 0.2% of events in which the histogram gradient was more than 0.3% was defined to be the lowest limit of the data, and the highest bin in which the histogram gradient was below −0.2% was defined as the upper limit of the data. The stained events were then gated to remove cell doublets that are identified as events with high fluorescence signal width. To this end, only events closer than 2.5 standard deviations of fluorescence width to the median of fluorescence width were taken as singlets. If as a result of selection of stained and singlet events less than 5,000 events remained, the strain was discarded from the analysis and re-measured. Distribution of cell cycle stages was determined by the method of Dean & Jett (1974). All results were manually verified to correct occasional incorrect determination of the cell cycle distribution. Cell size was estimated by looking at the width of the forward scatter signal, the parameter that exhibited the best correlation with the results of previous screens.

Both cell cycle and cell size measurements exhibited a considerable variability between days and between plates. We thus normalized the mean and standard deviations of the reported parameters so that every screened 96-well plate had the same mean and standard deviation.

### Time-lapse microscopy

Cells were pre-grown for around 24 h in SC medium to $OD_{600}$ of about 0.5. The carbon sources used were as follows: 2% glucose, 2% galactose, 0.05% glucose and 2% raffinose. The cells were then prepared for imaging on agar pads in 96-well plate with the respective SC as previously described (Bean *et al*, 2006). We observed growth of microcolonies at 30°C using fully automated Olympus IX71 inverted microscope equipped with a motorized XY and Z stage, external excitation and emission filter wheels (Prior) and an IR-based fast laser autofocus (Paran *et al*, 2007) using 60× air objective. Fluorescent proteins were detected using EXFO X-Cite light source at 12.5% intensity and Chroma 89021 mCherry/GFP ET filter set. Exposure time for the detection of eGFP and mCherry was 120 ms. Imaging was done by cooled EMCCD camera (Andor). The microscopic setup allowed simultaneous imaging of 60 fields of view for 6 h. Bright field, red and green fluorescence images were collected every 3 min.

### Image analysis

Identification and tracking of dividing cells was performed by custom-written software in MATLAB (Mathworks). Movies were analyzed from the end to the beginning, segmenting cells only in the last image and then tracking them to the first image. Nuclear marker facilitated the initial tracking and segmentation. Nuclear separation was identified by appearance of the nuclear marker in the daughter cell. Cell birth, defined by the bud neck disappearance, was identified as a significant decrease in the intensity of the bud neck marker in proximity (up to 30 min) to the nuclear separation. Cell volume was estimated from the bright field images assuming that the yeast cells are prolate spheroids (Lord & Wheals, 1981). Our results remain qualitatively the same when considering the area of the cell instead of the volume.

### Data analysis

To determine the relative time and size offset of the mutants relative to the wild-type, we found the overlap of the intervals containing 80% of the mutant and the wild-type. Then, the interval was split into 10 equally spaced bins, and the medians of the G1 times and volume increases in G1 for both strains were calculated for the cells in each bin. Relative volume increase $\Delta V$ and length of G1 were the average differences in the medians calculated over all size bins. To determine whether the calculated offsets were significantly different from zero, we applied Wilcoxon's rank-sum test for the data in each bin and calculated the *P*-value for the difference of medians in this well. We then united the *P*-values between the bins using Fisher's method.

To prevent artefacts stemming from the finite lengths of our movies, we considered only cells born at least 100 min prior to the end of the move in our analyses. In the analysis of Fig 7, we

pooled together data from WT, *whi5*, *whi3*, *kap122*, *ste20*, *lte1*, *gpr1* and *gpa2* which were chosen as strains with cell cycle of a similar length to the wild-type. To avoid effects from different amount of measured cells in the strains, we randomly picked 300 cells from each strain.

**Competition experiment**

Cells were grown to stationery phase in SC medium overnight (OD ~10). The unlabeled tested and wild-type strain expressing mCherry under constitutive promoter were then co-incubated in SC at 30°C. The initial OD was set to ~0.05, and the WT initial frequency was ~50% from the total population. Generation times were calculated from the dilution factor. Frequencies of labeled versus unlabeled cells were measured by flow cytometry. The cells were diluted once per day and reached stationary phase. A linear fit of the $\log_2$ for the WT frequency dynamics was used to calculate the slope for each competition assay. The relative fitness advantage was calculated from the slope divided by $\log_2$.

**Data availability**

The raw data of the flow cytometry screen are available in Flowrepository (http://flowrepository.org). The data of the microscopy screen are available upon request.

**Supplementary information** for this article is available online: http://msb.embopress.org

## Acknowledgements

We thank C. Boone and the Yeast Resource Center for plasmids and strains, Z. Kam for help with setting the microscopic system, A. Doncic for the initial development of the image analysis software, D. Ben-Zvi, G. Hornung, S. Levy, D. Rosin and Y. Voichek for discussions, and members of our laboratory for support and discussions. This work is supported by the European Research Council (IDEAS). N.B. is the incumbent of the Lorena Greenberg Scherzer Professorial chair.

## Author contributions

NB and IS conceived and designed the study. IS performed experiments and developed the image analysis software. NB and IS analyzed the data and wrote the manuscript.

## Conflict of interest

The authors declare that they have no conflict of interest.

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
