## [Review Process File · Molecular Systems Biology]

Systematic identification of cell size regulators in budding yeast

Ilya Soifer and Naama Barkai

Corresponding author: Naama Barkai, Weizmann Institute of Science

Review timeline:

Submission date:	06 April 2014
Editorial Decision:	11 May 2014
Revision received:	09 August 2014
Editorial Decision:	22 September 2014
Accepted:	15 October 2014

Editor: Thomas Lemberger

Transaction Report:

1st Editorial Decision

11 May 2014

Thank you again for submitting your work to Molecular Systems Biology. We have now heard back from the three referees who agreed to evaluate your manuscript. As you will see from the reports below, the referees find the topic of your study of potential interest. They raise, however, substantial concerns on your work, which, I am afraid to say, preclude its publication in its present form.

The reviewers appreciate that value of your dataset as a resource. They raise however quite serious concerns with regard to the interpretation of the data. As such they request additional experimentation and analyses to be performed, in particular to support the conclusions with regard to the impact of ribosomal genes, a much clearer and rigorous presentation of the results and a better discussion of the connection of this study with previous works in the field.

We would also kindly ask you to also include the data of the prescreen, either as stand alone dataset file or as 'source data file' directly linked to a specific figure panel (see also our instructions at <http://msb.embopress.org/authorguide#a3.4.3>). We would also strongly suggest to include a sub-section 'Data availability' at the end of the Materials & Methods section to list the datasets produced in this study.

If you feel you can satisfactorily deal with these points and those listed by the referees, you may wish to submit a revised version of your manuscript. Please attach a covering letter giving details of the way in which you have handled each of the points raised by the referees. A revised manuscript will be once again subject to review and you probably understand that we can give you no guarantee at this stage that the eventual outcome will be favorable.

Reviewer #1:

Summary

- Describe your understanding of the story

Examine genetic mechanisms of cell cycle progression, the interplay between cell size at birth, at initiation of division and volume growth rate in between, based on time-lapse single cell analysis of a large cohort of mutants. The study identified some additional size control genes and evaluated the role of ribosome components in the timing of initiation of cell division, a topic of some controversy. They also point to a size-dependent control point after initiation of cell division.

- What are the key conclusions: specific findings and concepts

They argue that growth rate is a key determinant of size at budding. They conclude that ribosome components are positive regulators of initiation of cell division, with a clear dichotomy between large and small ribosomal protein subunits. In this reviewer's opinion, the major strength of the manuscript is in the microscopic single-cell analysis of more than 500 mutants. This dataset provides a valuable resource to the community.

- What were the methodology and model system used in this study

They applied previously developed high resolution single-cell analysis of cell cycle progression in hundreds of mutants of *Saccharomyces cerevisiae*.

General remarks

- Are you convinced of the key conclusions?

Partly. Their pre-screen is of low value as described. The claims about the deterministic role of growth rate are not well supported. The different behavior of *rpl* vs *rps* mutants is not rigorous enough, and the evidence for a bypass mechanism in the budded phase is weak at best.

- Place the work in its context.

This is a more systematic approach to examine at single-cell resolution a far larger collection of mutants than previous studies. Regardless of the interpretation/conclusions from these measurements, if the data are presented in a detailed and easily accessible form, this study will make a significant contribution to the field.

- What is the nature of the advance (conceptual, technical, clinical)?

The nature of the advance is technical, in providing cell cycle parameters, such as volume growth in G1 and critical size data, for 500+ strains.

- How significant is the advance compared to previous knowledge?

The advance is in "degree" (i.e., larger dataset of mutant analyses for volume growth in G1 and critical size), not in "kind". Because of this "strength in numbers" the results do advance the field by extending prior conclusions that relied on narrower sampling.

- What audience will be interested in this study?

Cell biologists, primarily yeast cell cycle researchers, but others from other model systems as well.

Major points

-Specific criticisms related to key conclusions

1. In the opinion of this reviewer the analysis in Fig. 1 (the pre-screen) is unconvincing and poorly described, for the following reasons:

- a) Fig. 1A shows some examples (WT, *whi5*, *bck2*) of the fitted data from their flow cytometry. These fits are simply too good to be true. They should make available their raw flow data for proper evaluation and for future independent analyses.
- b) What is Fig. 1D showing? They seem to conclude that their FSC-based screen performed adequately in identifying the previously classified *whi* mutants based on actual cell size measurements. However, all the figure shows is that the majority of *whi* mutants they included here are on the left-of-median portion of their FSC distribution. This is not a strong endorsement of their pre-screen approach. The *whi* mutants were selected as the smallest 5% of the strains analyzed by Jorgensen et al, and they now seem to overlap with 50% of the FSC distribution. Finally, are only 26

whi mutants included in this analysis? If so, what about the rest small size mutants that Jorgensen et al found?

c) How is one supposed to read Fig. 1E? There is no information in the figure legend. There is a color-coded matrix of some kind, but what is on the horizontal axis? What are the actual correlations (real numbers), and what statistical tests were used to derive them? I was not able to locate this information anywhere. This is very important because the authors routinely and throughout the text use "cell size" to describe FSC values (their study) and size calculations from microscopic images (their study and Ohya et al), but the gold standard in the field for cell size values are from electric particle analyses (used in Jorgensen et al). How these outputs match with each other is critical here. At least in the case of FSC (which was used in their pre-screen), other studies have indicated that FSC is a poor surrogate of cell size (see PLoS Genet. 2012;8(3):e1002590 and also see FEMS Microbiol Lett. 1994 Apr 1;117(2):225-9 for other microbes). Finally, from the Jorgensen et al dataset, which value did they use for comparisons, the mean or median?

d) The rationale for selecting the 591 mutants for their next steps is unclear. It seems they cherry-picked whatever seemed interesting from their screen and from previous studies, but no cutoffs for any of these decisions are given. Is that true? If so, what is the value of the pre-screen then?

2. There are some conclusions that are drawn from the putative effects/dependencies between growth rate and cell size that are not clear or convincing, or they are simply wrong.

a) How did they quantify volume growth rate, as an exponential function or as a linear one? The Ferrezuelo study they cite treated it as a linear function. Which model (and why) they use is critical, since exponential models would incorporate changes in size.

b) They claim that "slow growing cells grew less in G1 than faster growing cells born at the same size" and they point to various figures. But how does one reach this conclusion from Fig. 2D and E they refer to? The slopes of the lines look very similar.

c) What do they mean to show in table S5? They state that "the durations of G1 in different carbon sources are insignificantly or almost insignificantly different, while sizes at budding are different very significantly". But the duration of G1 in different media is most certainly different (longer G1 in poorer media). Sizes at budding but also sizes at birth are also very different, with reduced size at birth accounting for most of the lengthening of G1 in poorer media. As I also comment elsewhere, normalizing G1 duration against birth size (which I think is what the authors are doing) is fine for evaluating size control efficiency, but that doesn't mean that birth size alone is not regulated in the mutants they examine or in poor nutrients, and it certainly does not imply that G1 length differences are "insignificant". It seems far less biased to me to simply treat and report each of these variables (birth size, volume growth rate in G1, budding size) separate from each other first, calculate the total length of G1 (which is really what matters for acceleration or not of G1/S, in my opinion) and then try to derive any relationships.

d) They proclaim that "cells growing at different growth rates but born at the same size budded after the same time", in effect making growth rate a key determinant of critical size. I am not sure they can claim that from the type of analysis they show in Figs. 2F and G. These figures simply show that the smaller cells are when they are born, the longer they will stay in G1.

e) Their own analysis of mutants in later figures, in my opinion, argues against the broad conclusions they draw about the deterministic role of growth rate on budding size (see Fig. 4). Also in dataset/table 3, there are mutants with lower volume growth rate and larger size at budding (e.g., rps0b). How do they explain those? Overall, I do not understand most of their arguments on the role of growth rate on budding size, and the ones I think I understand I do not find them very convincing. Finally, if indeed growth rate is a key determinant of size at budding as they claim, why do they seem to delegate growth rate mutants to a less interesting status in later parts of the manuscript?

3. The categorization of the mutants shown in Fig. 4 should be better explained and illustrated.

a) The statistics used to classify the mutants in Fig. 4 are not shown (I could not find this information in materials and methods as it was referenced). Which test was used, and what were the p values for each of the 9 groups of mutants and the cutoffs used to place them in these 9 categories?

b) The color coding seems to match the colors of dataset 3. If true, where would the "blue" mutants fall? Shouldn't one expect the category with extended G1 and decreased birth size to be more populated based on their previous statements (it now seems to only have 10 mutants - or maybe I am not reading this correctly). Where would mutants such as tor1 and sch9 fall?

c) They state in the legend of fig 4 that mutants having shorter G1 but normal volume growth were not classified as negative regulators because they are all expected to be false positives. Would that include mutants that are born large, divide at normal size, hence they do not grow as much in G1

and have a shorter G1? Where would they place such a mutant, and why would such mutants deserve no interest? It appears that they are normalizing for birth size when they determine duration of G1 in the mutant categorization. Is that true? If so, why? Small birth size is an important physiological response (e.g., in poor nutrients birth size -more so than critical size- is reduced, accounting for the longer G1 and delay in initiation of cell division). Normalizing the G1 of various mutants against their birth size (if indeed this is what the authors are doing) introduces a significant bias.

4. The data in Fig 6 about the completely different effects in rpl vs rps mutants on cell size are very problematic. There is no evidence from previous studies that rps mutants as a group have increased size. The Jorgensen et al and the Zhang et al studies in 2002, relying on channelyzer data, show no such trend. If anything, the case is the opposite from what the authors state i.e., rps as a group have smaller overall size than wild type, although there are differences with rpls and within individual rps mutants. In my opinion, the conclusions presented are either a consequence of sample bias in their cherry-picking of mutants, or systematic errors in their assay.

5. Their evidence for a bypass mechanism that controls cell size in the budded phase is rather weak. They report a correlation of 0.24 (again, the type of statistical analysis they used is not clear), but this is hardly a strong support for the mechanism they propose. If I am not mistaken, the Di Talia et al 2007 study that used similar methodology, which they cite, found no evidence for size control in the budded phase. Finally, the whi3 and whi5 mutant analyses they show are indirect and they do not directly address the question whether there is a cryptic size control in the budded phase.

-Specify experiments or analyses required to demonstrate the conclusions

1. Overall, the relevant sections of their pre-screen as it is done and presented has very little value. In addition to their raw flow cytometry data, they should collate in a separate dataset all the relevant values from all these studies and theirs, side by side, and present actual numerical analyses of their correlations, so the readers can easily compare these studies and properly evaluate the authors' conclusions. After such an analysis, it is possible (perhaps even likely) that their arguments for using FSC as a pre-screen for cell size will not hold much water. In that scenario, the data will still be valuable to the field, as a side-by-side comparison of these different approaches to query "cell size". If that is the case, they should reformat this section not as an accurate pre-screen for their microscopic analysis that follows, but as a comparative analysis of methods that report on size, define how they selected their 591 mutants and move on to the more interesting parts of the manuscript.

2. Present independent and separate analyses of volume growth rate as a linear and as an exponential function, both for wild type cells in different nutrients and for their mutants, and draw appropriate conclusions. Also, they need to cite and correlate their study with others that reported on similar topics. Especially with their microscopic analysis, they need to refer to Kang et al (Integr. Biol., 2014, DOI: 10.1039/C4IB00054D). For their birth size measurements they need to correlate with the values reported in Truong et al (G3. 2013 Sep 4;3(9):1525-30).

3. The categorization of the mutants shown in Fig. 4 needs improvement. They need to give specific examples of mutants, explain the statistics, and better explain why they are classified as important or not for their conclusions.

4. To make the claims they make in Fig. 6 about large vs. small rp mutants, they need to include all the mutants, analyze them with repetitions and perform very robust statistics. As it stands, the reported larger than normal mean size and budding size for rps mutants is unsubstantiated and contradictory to previous independent studies. The onus is on the authors to convince the field otherwise.

5. In the absence of any new data, they would need to modify extensively their arguments about cryptic size control on the budded phase, in various places in the manuscript, including in the abstract. As it is now, they try to make too much out of a very weak result (a correlation of 0.24).

Minor points

-Easily addressable points

1. Please refer to the "Dataset" files as datasets, not as Supplementary tables, so it is easier to follow. Consistency in the labeling (in whichever way the authors prefer) in the text helps the reader.

2. Column C in Dataset 1 has no label (should be Normalized G1?)

3. In the subsection "Effect of translation and ribosomal biogenesis...", the first paragraph needs specific citations to back up their statements.

4. In the next paragraph from the one mentioned in the previous minor comment, they start with the statement: "The interpretation of these results was complicated by a general lack of single cell data making it hard to distinguish the direct from the indirect effects". Why is that so? I do not think that the controversy regarding the interpretation of ribosome mutant phenotypes has anything to do with single-cell vs. population-based data.

-Presentation and style

Style is fine

-Trivial mistakes

Check the text a little more carefully. When they reference something, it should be there. Also, articles (a/the) seem to be missing at several places.

For major revision, it is useful if you can provide a time estimation for the requested additional experiments/analyses.

I do not know. Depending on the throughput of their assay, 2-3 months might be enough.

Reviewer #2:

In this manuscript, the authors report the results of a high throughput screen for cell size mutants. Unlike previous such screens, the authors examine single cell correlations between the size of cells and budding and their size at birth as determined by automated segmentation of time-lapse movies. This allows the clean separation of size mutants that are small because they are giving birth to very small daughters rather than affecting the size control mechanism gating the G1/S transition. Thus, the screen is a clear improvement upon previous such efforts. The authors were able to uncover an interesting piece of biology in that mutations affecting the small subunit of the ribosome and translation initiation factors had a clear affect on G1/S control, while mutations in the large subunit, while they did affect population size and growth, did not affect G1/S control. This result might prove important in the determination of the molecular mechanism through which cell size is transduced to gate the G1/S transition, an important and ill-understood piece of biology. Following some minor revisions suggested below, this work should be published in MSB.

Figure 2: I found the density plots, especially when overlaid with 4 curves, quite difficult to use to compare different experiments and mutants. I think the presentation might be more clear if the authors plot the data using box plots after binning the data. Also, in this section the authors grow cells on 4 different environments, but none of them are really large perturbations in growth rate. If the authors want to claim support of a timer-type model, it would perhaps be more useful to examine cells growing much more slowly, such as on ethanol.

Figure 4: It would be better to plot Size in fl, which should be the same for everyone, than pixels, which are not.

Figure 5: Have the authors examined the recently published whi7 mutant from the Aldea lab (Mol. Cell)? It would be great to see how that mutant affects size control in the authors single-cell assay. Also, the y-axis of many panels in C have been clipped during some cutting and pasting to make the figure.

Figure 7: The WT data density plot is shown 3 times, which seems excessive especially. I have the same comments as for figure 2, where I think a more standard box-plot after binning by budding size would allow an easier comparison between mutants budding at the same size.

Last paragraph in the section titled 'Microscopic screen of...' refers to Fig 2F and G, when it should be Fig 3C, D.

First paragraph of the section 'Effect of translation and ribosomal...' is missing all references for the work described.

Reviewer #3:

The manuscript, "Systematic Identification of regulators of size in budding yeast using high throughput time-lapse microscopy" by Soifer and Barkai, describes a microscopic screen of the genome-wide deletion collection for mutants affecting cell size, largely during G1 phase, and their implications in terms of understanding cell size regulation. Importantly, the data are evaluated in terms of the relationship between size at birth, size at budding, growth during G1 and time in G1. That comparison showed that whereas the growth during G1 was dependent upon both the birth size and the growth rate, the length of G1 was only dependent upon birth size. Interestingly, this can account for the difference in budding size observed in different carbon sources. Comparing the effect of mutants on the relationship between G1 length and birth size, led to the classification of genes based upon their negative and positive effect on the size at bud emergence. Among those genes in which mutation causes an advancement in budding were several known to encode negative regulators of Start along with several for which such a role had not been previously described. Similarly, mutations in genes encoding known positive regulators of Start were observed to delay budding both temporally and in terms of growth during G1. However, when ribosomal protein genes were classified based upon their effect on Start, surprisingly, mutations in those encoding large subunit constituents advanced Start, whereas mutations in those encoding small subunit constituents delayed it. The authors conclude that elements of the large subunit appear to act as negative regulators of Start because diminished protein elongation limits cell growth during G1 and, therefore, leads to bud initiation at a small cell size, whereas those encoding elements of the small subunit appear as positive regulators because the cell cycle is positively regulated by translation initiation. Finally, the study shows that when cells pass Start at a small cell size, an additional size control mechanism is revealed late in the cell cycle.

This is a carefully executed single cell analysis of cell size control that attempts to shed some of the constraints of previous screens to discover new size control elements and uncover new modes of regulation of cell size. Furthermore, this study preselected a relatively larger set of mutants for analysis by single cell microscopy than had prior studies using an improved flow cytometric approach. That said, the novelty of the findings is relatively limited. The screen has revealed several genes not previously recognized to play a role in determination of size and uncovered an apparent bifurcation between the effects of mutation of large and small ribosomal protein genes on cell size at budding. Unfortunately, they have not further characterized the roles of these genes so it remains unclear whether they are direct regulators of cell size or affect the process indirectly. In addition, the authors neglect to tightly relate their work to the relatively extensive literature on cell size regulation based upon both population and single cell analysis. For example, they only fleetingly refer to models that distinguish between "timers" and "sizers", well accepted mechanistic models for size control, and otherwise discard commonly understood terminology, such as Start and bud emergence in lieu of the more ambiguous G1/S. Furthermore, although the behavior of known size control mutants appears to conform expectations, it is unclear precisely how their analysis relates to that in recent studies. This is important because previous studies seem to have established the independence of cell size determination from the birth size, whereas this study argues that birth size is a major determinant of size at budding. Finally, there are numerous unclear or poorly constructed arguments and many grammatical errors throughout the text, especially in the discussion. These issues are detailed below, along with a number of other issues, and will need to be addressed prior to publication of this work.

Specific points:

1. These authors express surprise that the effect of cell size at birth on the length of G1 (time from birth to budding) is independent of growth rate. However, Ferrezuelo et al, 2012 (Figure 3) show little correlation between the size at birth and the time spent in pre-start G1 phase or of the effect of carbon source on that relationship. Instead, they and di Talia et al, 2007 show a strong correlation between growth rate and size at Start. It is unclear whether this is a discrepancy between the results or, rather, one of interpretation. Although there is a difference between the studies in the cellular events measured (see below), it seems doubtful that this is sufficient to explain any discrepancy.
2. The authors have chosen to measure septin ring dissolution and assembly as indicators of G1 entry and exit. Other recent studies have chosen to measure Whi5 exit from the nucleus as the first

indication of cell cycle commitment or Start. The authors should discuss their results in the context of those differences. For example, is it possible that the cell size phenotype of some mutants a consequence of defects in the timing of septin ring formation rather than execution of Start? In addition, the difference in the phenotype measured raises a problem with terminology. This study does not formally measure cell size at Start but it also does not measure cell size at G1/S, the term used by the authors. S phase may coincide with septin ring formation under some growth conditions, but certainly not under all or in all mutants. It is probably best to refer to this as bud emergence.

3. The authors should compare and contrast their model with current models for size determination. Do these findings indicate that the cell size at budding is primarily determined by a "sizer" or "timer" mechanism or is instead explained by some other mechanism.

4. There is a relatively extensive literature concerning the effect of mutants in ribosomal protein genes and ribosome biogenesis genes as positive and negative regulators of Start. There are mutants in translation initiation factors that would seem to support a positive role for translation initiation in the regulation of Start (*cdc68*, for example). The authors should discuss whether studies or the effects of inhibitors of translation initiation and elongation are consistent with their observations and, perhaps, test the effect of such inhibitors on cell size in the context of their experiments.

5. Discussion: The section "Role of protein synthesis..." is quite confusing. First, the two models for the role of protein synthesis in the regulation of cell size presented by the authors both posit a positive role for protein synthesis but the second model has an additional condition, an opposing role for ribosome biogenesis. The same paragraph then goes on to provide two pieces of evidence in support of "this" model. Which model are they referring to? Presumably the second model. That should be clarified.

Second, the next paragraph goes on to argue that their data supports the first model, despite the fact that they have just provided their own data in support of the second model. Although I understand the arguments that are made, the construction of these two paragraphs makes the assignment of the various arguments to different models ambiguous.

6. The existence of a second size control point during G2/M that is revealed when the budding size is small has been previously reported.

Minor points:

1. Should cite recent paper showing that retention of *Cln3* at the ER is dependent upon *Srl3/Whi7* (Aldea lab).
2. Regulation of Start by regulators of ribosome biogenesis (*Sfp1*, etc.) should be mentioned in the introduction.
3. Page 3, line 4: Should read "The quantitative...." rather than "Quantitative...."
4. Page 6, paragraphs 2 and 3: The figure numbering is incorrect. Figure 2 F and G should be Figure 3 C and D. Figure 4 does not have A and B components.
5. Page 7: Should mention other reported functions of *Whi3* (Aldea lab).
6. Page 7: Paragraph discussing relevance of the connection between *Rsr1* and *Lte1* is highly speculative and, if it belongs anywhere, it should be in the discussion.
7. Page 8: Numerous studies are mentioned without reference in the first paragraph of the section "Effect of translation...." These references should be cited. Also, the last paragraph should start with "A recent...."
8. Page 8: It may be appropriate to cite and discuss Thapa et al, 2013 which catalogs the effects of RP mutations on cell morphology, cell cycle distribution, etc.
9. It would be helpful to distinguish between cell "growth" and "proliferation" throughout the manuscript.
10. Page 9, paragraph 1 of second section: The third sentence is confusing. "...alternatively, some process starts at the beginning of the cell cycle and completes until the cell undergoes division." Please clarify.
11. Page 11, paragraph 2: "...five hundreds..." should be "...five hundred..." "Large proportion..." should read "A large proportion..."
12. Page 11, paragraph 4: "...longer G1 that..." should read "...longer G1 than..."
13. Last sentence of Discussion: "...size homeostasis at the conditions of an exponential growth..." should presumably read "...size homeostasis under conditions of exponential growth."

14. Page 12, last paragraph: The 5th sentence refers to a second model but there is no reference to two models of size control earlier in the section. Is this a reference to models in the previous section. If so, it should be made clearer.
15. There are many other typographical and grammatical errors throughout. The paper should be carefully edited.

1st Revision - authors' response

09 August 2014

Reviewer #1:

Summary

- Describe your understanding of the story

Examine genetic mechanisms of cell cycle progression, the interplay between cell size at birth, at initiation of division and volume growth rate in between, based on time-lapse single cell analysis of a large cohort of mutants. The study identified some additional size control genes and evaluated the role of ribosome components in the timing of initiation of cell division, a topic of some controversy. They also point to a size-dependent control point after initiation of cell division.

- What are the key conclusions: specific findings and concepts

They argue that growth rate is a key determinant of size at budding. They conclude that ribosome components are positive regulators of initiation of cell division, with a clear dichotomy between large and small ribosomal protein subunits. In this reviewer's opinion, the major strength of the manuscript is in the microscopic single-cell analysis of more than 500 mutants. This dataset provides a valuable resource to the community.

We thank the reviewer for the positive evaluation of our data. Concerning the first point, we would like to mention that although we show that growth rate strongly affects the size of budding, we do not in fact argue that it is a part of the size control mechanism. What we show is that the most reliable measure of size control is actually G1 duration, which depends on the birth size (and therefore compensates for variation in this size by extending G1 in small cells) but does not depend on growth rate. The mutants that we focus on in the context of this work are therefore those that change the dependency between the G1 length and the birth size and thus their effect on size cannot be explained only by their effect on the growth rate. We chose to focus on these mutants since their effect on size control is most likely direct and not through growth rate. We rewrote a large part of the manuscript to better clarify this central point.

- What were the methodology and model system used in this study

*They applied previously developed high resolution single-cell analysis of cell cycle progression in hundreds of mutants of *Saccharomyces cerevisiae*.*

General remarks

- Are you convinced of the key conclusions?

*Partly. Their pre-screen is of low value as described. The claims about the deterministic role of growth rate are not well supported. The different behavior of *rpl* vs *rps* mutants is not rigorous enough, and the evidence for a bypass mechanism in the budded phase is weak at best.*

We hope that our specific answers below are now more convincing:

1. We now provide more results of the pre-screen and made all the data available in a public database. In the text, we significantly reduced emphasis on the pre-screen.
2. We note that the fact that growth/proliferation rate affects size had been shown by many studies. Our aim in this study was to identify mutants whose effect on size control cannot be explained only by their effect on proliferation rate.

3. We explain below why we trust our conclusions regarding the differential role of *rpl* vs. *rps*.

- *Place the work in its context.*

This is a more systematic approach to examine at single-cell resolution a far larger collection of mutants than previous studies. Regardless of the interpretation/conclusions from these measurements, if the data are presented in a detailed and easily accessible form, this study will make a significant contribution to the field.

We thank the reviewer for his/her support.

- *What is the nature of the advance (conceptual, technical, clinical)?*

The nature of the advance is technical, in providing cell cycle parameters, such as volume growth in G1 and critical size data, for 500+ strains.

- *How significant is the advance compared to previous knowledge?*

The advance is in "degree" (i.e., larger dataset of mutant analyses for volume growth in G1 and critical size), not in "kind". Because of this "strength in numbers" the results do advance the field by extending prior conclusions that relied on narrower sampling.

- *What audience will be interested in this study?*

Cell biologists, primarily yeast cell cycle researchers, but others from other model systems as well.

Major points

-Specific criticisms related to key conclusions

1. In the opinion of this reviewer the analysis in Fig. 1 (the pre-screen) is unconvincing and poorly described, for the following reasons:

*a) Fig. 1A shows some examples (WT, *whi5*, *bck2*) of the fitted data from their flow cytometry. These fits are simply too good to be true. They should make available their raw flow data for proper evaluation and for future independent analyses.*

Our flow-cytometry data is indeed of a very high quality. A similar flow-cytometry data was reported by us before for S phase mutants (Koren *et al*, 2010) which were of a similar high quality. We improved typical protocols, using SYBR green for DNA staining instead of SYTOX green. This, as pointed by us and others results in a significantly lower variance of the peaks and better resolution (Fortuna *et al*, 2001; Haase & Reed, 2002). We replaced the plots on Figure 1A with histograms and fits with a higher number of bins to make it easier to appreciate the quality of the flow cytometry data.

All the flow cytometry data were uploaded to the Flowrepository, a public repository for flow cytometry data. The data is currently open to the editor and the referees, and will be available to everyone upon publication. Please use the following access keys for the data (the whole pre-screen was split into eleven chunks because of the database requirements):

Dataset	Link
pre-screen 1/11	http://flowrepository.org/id/RvFrhTFfc2IQ8Red1LbnLUuP3xQNr8Jp9j90POTUNDyqO02mdQ83YpUEX7UroFa6
pre-screen 2/11	http://flowrepository.org/id/RvFrKt7Foi536vAfzAwsYxPcLf41NH66bLJ2Llc15Gvzwuo3eWK8baXLxj7GbouC
pre-screen 3/11	http://flowrepository.org/id/RvFrk75qGZU446lLNFFVJycEoWsgCGdDz7EGGnddKtksE0NVSmCmU45duOAcwpc3
pre-screen 4/11	http://flowrepository.org/id/RvFr4tb3psbHfBn1twVFSI0tAZ46BaHHQ95GtlMaOJLdj2Fiwo1iiBsIOqxMT9R7
pre-screen 5/11	http://flowrepository.org/id/RvFr8MJMZ1zZjrii0ykrxQ2InOLPGQoZIFW1xEjwOU8ZzMa1w5xZ6Bge5Cf2OwiA
pre-screen 6/11	http://flowrepository.org/id/RvFr8eUPbrafrBWI9J3fe9ocL2FYlCpnphPIUHeF9CUp6oTYhlQcIjj7CSWqX9Cr
pre-screen 7/11	http://flowrepository.org/id/RvFrusAeuqIRBSrB216eklqfcM3xcHKmfbKIEEvQIMxwsfjAucNFqplnD3rOOk3b
pre-screen 8/11	http://flowrepository.org/id/RvFr3kb8Ikjam2abCKb618dJPSH2nPDW7fy9tYumUAV

	T4FfT47GCV5bH4Jglphh2
pre-screen 9/11	http://flowrepository.org/id/RvFrikP3dPzAdisqrXiB12Mnt0W39ffQb8iXncv0g5911C9J7ws3fTNEngadR9sq
pre-screen 10/11	http://flowrepository.org/id/RvFrVY49rJZOrGXqjG7nP0b4y5mxQYoBwjleRkrPpCKMa8PUeC32KjJNMZei5IHu
pre-screen 11/11	http://flowrepository.org/id/RvFrUvz2z2xCmzfBIBi7GPEHwdvC0kCLLZuYtHmnTQ5100UE1DI0Fn2zaQZ7JjOH
Repeat extended G2	http://flowrepository.org/id/RvFredj5gp4vGmJsHGOxPCKKYfr0CZUCJhWSc10o8JVQXr901GUpu2DgrhOsHceS
Repeat extended G1	http://flowrepository.org/id/RvFr0wgMeq07R1A4pIEG5smznYLGyG1TCK6RDISAprnfwLEMjwUpw6sziyCeGzt
Repeat extended S	http://flowrepository.org/id/RvFrBww03baWU9RYGYIs6KjOggySz2QwAQttpDT9iB1EPBeilZKPOW2iUDPvi5Kc
Repeat small size	http://flowrepository.org/id/RvFr0vXV5ihXKKcs3YsIdQ9fJf33rcHuuxmmfEDNuJ2JpH1wrxMIQj1JBdInLxtx
Repeat large size	http://flowrepository.org/id/RvFrJTg7z6qNG5AZsY71CvgTzutXHbMDvffP5gULuFbyZlIwsbyhTQr5UYsZwUAI
Repeat special interest	http://flowrepository.org/id/RvFrTVbuDQ7pdrDUiHNGs3pRUbumulu1UgZpSZiLtSE02i575Dfw6C0rQmO0XR92C

b) What is Fig. 1D showing? They seem to conclude that their FSC-based screen performed adequately in identifying the previously classified whi mutants based on actual cell size measurements. However, all the figure shows is that the majority of whi mutants they included here are on the left-of-median portion of their FSC distribution. This is not a strong endorsement of their pre-screen approach. The whi mutants were selected as the smallest 5% of the strains analyzed by Jorgensen et al, and they now seem to overlap with 50% of the FSC distribution. Finally, are only 26 whi mutants included in this analysis? If so, what about the rest small size mutants that Jorgensen et al found?

The FACS pre-screen was designed for defining new candidates to complement candidates defined by several previous static screens to be analyzed at more depth using the live microscopy. Those high-throughput static screens are rather noisy and depend on the precise conditions where measurements were taken, as indicated by the low correlation between existing screens reported by several groups. Our screen shows a similar overlap with previous screens, as found between those previous screens themselves, and this is what is shown in Figure 1D. For the actual microscopy screen, we combined the data from our screen with data from previous screens as well as from other functional experiments, in order to optimize our ability to detect real size regulators (We added detailed explanation for the selection of strains for the screen in the Expanded View Text and Dataset E2)

We added a new subplot to Figure 1D comparing average sizes of small, normal and large size mutants in Jorgensen et al in our screen. Half of the small strains identified by Jorgensen et al were in the lowest 20% strains in our screen and 75% were in the lower half of the size distribution. Over half of the large strains are in the highest 20% sizes in our screen. Overall the mean size of the small mutants found by Jorgensen et al was significantly below the mean size of all mutants ($p\text{-val} < 10^{-26}$) and the mean size of the large mutants found by Jorgensen et al is significantly higher ($p\text{-val} < 10^{-100}$). The remaining differences could be a result of different measurement conditions (e.g. we consistently do not observe small sizes of mitochondrial mutants, likely reflecting the lower demand for respiration in our 96-well growth conditions) or measurement noise. As we were aware of significant noise in size estimation, 25% of small and large strains were re-measured as well as some strains that were suspect to have an interesting phenotype and only the strains that showed a reproducible phenotype were subject to a microscopic screen.

c) How is one supposed to read Fig. 1E? There is no information in the figure legend. There is a color-coded matrix of some kind, but what is on the horizontal axis? What are the actual correlations (real numbers), and what statistical tests were used to derive them? I was not able to locate this information anywhere. This is very important because the authors routinely and throughout the text use "cell size" to describe FSC values (their study) and size calculations from microscopic images (their study and Ohya et al), but the gold standard in the field for cell size values are from electric particle analyses (used in Jorgensen et al). How these outputs match with each other is critical here. At least in the case of FSC (which was used in their pre-screen), other

studies have indicated that FSC is a poor surrogate of cell size (see PLoS Genet. 2012;8(3):e1002590 and also see FEMS Microbiol Lett. 1994 Apr 1;117(2):225-9 for other microbes). Finally, from the Jorgensen et al dataset, which value did they use for comparisons, the mean or median?

The figure was edited and is now better annotated. We apologize for previous omissions. We now show the value of Pearson correlation coefficient obtained for comparisons of the estimated sizes and G1 lengths from different screens. For comparison with Jorgensen et al we used their median cell volume.

We agree with the reviewer that FSC has its drawbacks but for us it was a useful parameter that we could obtain simultaneously with measuring cell cycle distribution. We now take care not to use the term “cell size” but “FSC”. As figure 1E shows, the correlation between cell sizes measured in every pair of screens was not very high, which is probably a result of either technical noise and/or different measurement conditions. Results from our screens overlap with previous screens to the same extent as those previous screens overlap between themselves.

d) The rationale for selecting the 591 mutants for their next steps is unclear. It seems they cherry-picked whatever seemed interesting from their screen and from previous studies, but no cutoffs for any of these decisions are given. Is that true? If so, what is the value of the pre-screen then?

We selected candidates from our screen in a rigorous manner by choosing strains that were significantly smaller/larger than expected from their G1 duration and behave in a similar way in the repeats. For the estimation of their size we used either our forward-scatter based estimates or the estimates by Jorgensen et al by electronic volume measurement. We added the description of the selection of the strains for the microscopic screen to the Expanded View Text (section 4) and see below.

2. There are some conclusions that are drawn from the putative effects/dependencies between growth rate and cell size that are not clear or convincing, or they are simply wrong.

We thank the reviewer for this comment which made us realize that our axis labeling in some of the figures is confusing; we do not in fact measure growth rate, but simply report the overall increase in volume from the time of birth until budding (volume is estimated from our live imaging). Growth rate is thus not a critical factor in our analysis. All relevant legends were now corrected (changed ‘volume growth’ to ‘DV in G1’ (DV= \log - volume(bud)- \log -volume(birth)))

a) How did they quantify volume growth rate, as an exponential function or as a linear one? The Ferrezuelo study they cite treated it as a linear function. Which model (and why) they use is critical, since exponential models would incorporate changes in size.

As mentioned above, our analysis does not require a certain model of the volume growth: since we compare the total change in volume (or the duration of G1) for cells that were born at the same size. Therefore, differences in the growth rates coming from the different birth sizes (if growth rate is exponential) do not affect this comparison.

We note in passing that our data on volume growth is consistent with exponential growth. We estimated growth rate using both linear and exponential approximations and found that when exponential approximation was used, the specific growth rate was independent of birth size. In contrast, when linear approximation was used, the fitted growth rate was strongly correlated with birth size. This supports the exponential growth model. We added this result to the SI for the interested reader (Figure E3A,B).

b) They claim that "slow growing cells grew less in G1 than faster growing cells born at the same size" and they point to various figures. But how does one reach this conclusion from Fig. 2D and E they refer to? The slopes of the lines look very similar.

We apologize: as mentioned above, the axes labeling were confusing. What we show is the total change in volume between birth and budding. The different curves correspond to cells growing in different media (hence at different growth rate). So what should be compared is not the slopes of the

curves (which are approximately the same, as pointed by the reviewer) but the fact that the different curves are shifted. Cells that are born with the same size, add more or less of a volume, depending on their growth media: in media where they grow slow they add less volume, while in media where they grow fast they add more volume.

c) What do they mean to show in table S5? They state that "the durations of G1 in different carbon sources are insignificantly or almost insignificantly different, while sizes at budding are different very significantly". But the duration of G1 in different media is most certainly different (longer G1 in poorer media). Sizes at budding but also sizes at birth are also very different, with reduced size at birth accounting for most of the lengthening of G1 in poorer media.

Our formulation was confusing and we apologize for that. It is well known that size at budding is different in different media and our results clearly agree with that. Our main new observation here is that *when comparing cells of the same birth size, the duration of G1 does not depend on cell growth rate*. Thus, the duration of G1 depends on birth size ***in a way that is independent of the external nutrient and the associated growth rate***. This finding is quite surprising in fact: it means that if, by chance, cells in two different media are born at the same size they will remain in G1 for the same durations (and at this time, will add more or less of a volume, depending on their growth rate). The difference in the average G1 duration between cells growing at different media comes from the fact that they are born at different average sizes: slow growing buds add less volume, are born smaller and therefore spend longer time in G1.

Most models of size control do not make this prediction, but assume that size difference is due to e.g. different size threshold a budding. Our observation that the functional form $T(V_{G1})$ remains constant allows attributing size differences in different media simply to changes in the rate of volume increase, but not changes in size control.

We note that our main motivation for comparing cells growing in different media, and for examining properties that remain invariant to cell growth rate, was to understand how to compare size control in mutants that grow at different rates. We wanted to identify mutants that affect cell size in a way that could not be explained just by their effect on the growth rate. In principle, different models of size control would make different predictions about how to normalize for differences in growth rate. In the checkpoint model, for example, if the threshold does not depend on growth rate, no normalization is necessary. By contrast, if the threshold depends on growth rate, this dependency should be normalized for. Other models would make different predictions. We therefore decided to employ an empirical approach to examine the effect of changing growth rate, in wild type cells where we know that size control is at work. Our results suggest that the dependency of G1 duration on birth size is the most robust measure that shows little dependency on growth rate, and we therefore used it for comparing size control in different mutants.

As the table caption was confusing, we changed it and it now says (note that the table is now table E2):

Table E2. p-value for the two null hypotheses: (1) if (birth-size normalized) G1 duration (T_{G1}) is equal between two conditions (i and j) and (2) if the (birth size normalized) volume increase during G1 $v_s - v_b$ is equal between the two conditions. The test was performed as described in Materials and methods. Note that the differences in the durations of G1 for cells born at the same size in different carbon sources are insignificant (or almost insignificant), while differences in their volume increases in G1 are highly significant.

As I also comment elsewhere, normalizing G1 duration against birth size (which I think is what the authors are doing) is fine for evaluating size control efficiency, but that doesn't mean that birth size alone is not regulated in the mutants they examine or in poor nutrients, and it certainly does not imply that G1 length differences are "insignificant".

We corrected our writing: we did not mean that the changes in G1 length or in size are insignificant. What we referred to as insignificant is ***the difference in G1 duration for cells born at the same size***. This is now better explained both in the text and in the caption.

It seems far less biased to me to simply treat and report each of these variables (birth size, volume growth rate in G1, budding size) separate from each other first, calculate the total length of G1 (which is really what matters for acceleration or not of G1/S, in my opinion) and then try to derive any relationships.

We hope that the new figures/ axes labels and explanations of the different relationships make our claims easier to understand. We agree with the reviewer that the average length of G1 is the main factor that determines whether the G1/S transition is accelerated. However, defining the parameters of size-control in different mutants, required distinguishing between birth size-dependent effects on the length of G1 (small born cells have longer G1, so G1 could be extended or shortened just because cells are born small/large due to e.g. slow growth or extended budded period) and size independent effect on the length of G1. Our method, which correlates birth size to the length of G1, is therefore able to distinguish between these two effects.

d) They proclaim that "cells growing at different growth rates but born at the same size budded after the same time", in effect making growth rate a key determinant of critical size. I am not sure they can claim that from the type of analysis they show in Figs. 2F and G. These figures simply show that the smaller cells are when they are born, the longer they will stay in G1.

Figure 2F shows the time spent in G1 as a function of cell size at birth for cells growing in different media (different media are shown in the different colors). The key point is that this dependency is virtually the same for all media, although growth rates in those media are different: cells that are born at a given size, spend the same time in G1, independently of whether they are e.g. growing in glucose (fast growth), or galactose (slower growth).

In our previous version, the different curves in the figure were not clearly emphasized (as noted also by the other reviewers) and we have now modified the figures to make them clearer to understand. We also changed the writing to emphasize more what we actually see (time in G1 for cells born at a given volume is independent of the growth media) and what we infer from that (growth rate does not affect the time in G1 for cells born at the same volume). Another indication that growth rate does not affect the time cells spend in G1 comes when looking more directly for correlation between the size at budding and the growth rate. Cells that are born at the same size but grow at different rates will bud at different sizes (Figure E3C), suggesting that the rapidly/slowly growing cells are unable to compensate for the difference in growth rates.

In a more general sense, in terms of critical size model, the finding that G1 duration is independent of cell growth rate may be interpreted in different ways. In the framework of the checkpoint model, It means that critical size depends on growth rate in a fine-tuned way that makes G1 duration invariant to the rate of volume increase. Alternatively, it may suggest an alternative model of size control that is different from the critical size concept according to which cells do not initiate budding when reaching a critical size but rather, that cells control the duration of G1 based on their birth size, largely independently of growth rate. Therefore, in this interpretation, growth rate does not affect aspects of the size control mechanism, but simply interprets the main control mechanisms (that function by modulating G1 duration) into cell size. We note that while we favor the later possibility for reasons of Occam's Razor, which interpretation is correct is not relevant to our study, as our goal is to simply normalize mutants growing at different rates, which is the same independent of the underlying model.

e) Their own analysis of mutants in later figures, in my opinion, argues against the broad conclusions they draw about the deterministic role of growth rate on budding size (see Fig. 4). Also in dataset/table 3, there are mutants with lower volume growth rate and larger size at budding (e.g., rps0b). How do they explain those? Overall, I do not understand most of their arguments on the role of growth rate on budding size, and the ones I think I understand I do not find them very convincing.

We hope that our answers above now make our point clearer: the growth rate effect on the size at budding is not a direct indicator of size control. Rather, what remains invariant to growth rate (and can therefore be used to compare different mutants) is the (birth-size normalized) G1 duration. We indeed see many mutants (32) that show a change in budding size, but this change can be explained only by the reduced growth rate, while their (birth-size normalized) G1 duration remains the same as

that of wild-type. These mutants affect cell size by changing the growth rate, while the basic size control mechanism remains intact.

Other mutants, like the mentioned *rps0a*, do affect size control in the sense that their (birth-size normalized) G1 duration differs from that of wild type. For *rps0a*, G1 duration is significantly extended (even given their smaller birth size) and this duration over-compensates for their slower volume growth leading to an increase in budding size. Thus we classify *RPS0A* as a positive regulator

Finally, if indeed growth rate is a key determinant of size at budding as they claim, why do they seem to delegate growth rate mutants to a less interesting status in later parts of the manuscript?

Since these mutants do not perturb the (birth-size normalized) G1 duration, we do not assign them a role in the size-compensation mechanism itself, as explained above. While these mutants would be interesting in the study of cellular processes that set the growth rate, we wanted to focus on the factors that regulate size control.

3. The categorization of the mutants shown in Fig. 4 should be better explained and illustrated. a) The statistics used to classify the mutants in Fig. 4 are not shown (I could not find this information in materials and methods as it was referenced). Which test was used, and what were the p values for each of the 9 groups of mutants and the cutoffs used to place them in these 9 categories?

We now explain the statistics that was used both in the main text and in the new Figure E4. This is how we describe our statistical test in the Methods:

To determine the relative time and size offset of the mutants relative to the wild type, we found the overlap of the intervals containing 80% of the mutant and the wild type. Then the interval was split into 10 equally spaced bins and the medians of the G1 times and volume increases in G1 for both strains were calculated for the cells in each bin. Relative volume increase DV and length of G1 were the average differences in the medians calculated over all size bins. To calculate if the calculated offsets were significantly different from zero we applied Wilcoxon ranksum test for the data in each bin and calculated the p-value for the difference of medians in this well. We then united the p-values between the bins using Fischer's method.

And this is how we define the cutoffs in the Results:

This classification was done by dividing the cells into evenly spaced bins according to their birth size, calculating the average G1 duration in each bin and the average volume increase. These values were compared to the corresponding wild type values and p-value for the difference was calculated (see Materials and methods and Figure E4). Thus, we asked whether cells born at a given birth size spent longer/shorter time in G1 (or grow more/less) compared to wild-type cells born at the same size. About two-fifths (197) of the strains showed statistically significant (p-value<0.001) difference from the wild type (Dataset E3).

*b) The color coding seems to match the colors of dataset 3. If true, where would the "blue" mutants fall? Shouldn't one expect the category with extended G1 and decreased birth size to be more populated based on their previous statements (it now seems to only have 10 mutants - or maybe I am not reading this correctly). Where would mutants such as *tor1* and *sch9* fall?*

The color coding did not precisely match Figure 4 and we apologize for that. This was now fixed and we changed the labeling of the Dataset E3 to match Figure 4 exactly.

The category of strains with for which the average G1 is extended and the average birth size decreases is indeed highly populated and includes most of the strains showing a small size (32 out of 52 small strains belong to it). *tor1* is one example for such a mutant. We note that for those mutants, the increase in average G1 is due to a reduced birth size. When comparing cells born at a given size, however, this difference disappears and the mutant cells show the same G1 length as wild type cells. Hence in the table, those mutants are assigned an unaltered (birth-size normalized) G1 duration.

The category that contains ten mutants is different: in this class of mutants, G1 is extended also when normalized for birth size. That is: cells of this mutant have longer G1 relative to wild type cells even if born at the size. In addition, those cells grow less in volume than one would expect from the wild type cells born at the same size. Most of these mutants have a very slow growth rate. One of these strains was *sfp1* (despite repeated attempts we were not able to introduce fluorescent markers into *sch9* while maintaining its size phenotype, consistent with its propensity to accumulate compensatory mutations (Jorgensen *et al*, 2004)). We do not expect this category to be particularly populated.

We note that based on the flow of our analysis, those mutants are expected to be associated with perturbed size control. However, we are careful in making this assessment, because of their very slow growth rate, which falls out of the growth rate intervals for which we observed an independency of G1 duration on growth rate. Fluctuations in growth rate of individual cells in rich media rarely brings them to this slow growth rate regime, and the only media we tested that approached those slow growth rates is glycerol. Glycerol is quite different from the other carbon sources we examined (it is non-fermentative) and for which we see as somewhat different dependency of G1 duration on birth size. Since we do not know if this dependency at small growth rate can be extended to mutants growing in rich media, we decided to leave those stains unclassified.

c) They state in the legend of fig 4 that mutants having shorter G1 but normal volume growth were not classified as negative regulators because they are all expected to be false positives. Would that include mutants that are born large, divide at normal size, hence they do not grow as much in G1 and have a shorter G1?

No, the mutants classified as false positives are different: they grow as much as wild type (add the same volume, not less), but do it a shorter time. This seems to us to be highly unlikely. Based on wild-type repeats, we estimated the number of false-positive expected this category, and this number is in fact higher than the number of mutant that are actually classified to this category (5 mutants). This led us to conclude that these cases are false positives.

Where would they place such a mutant, and why would such mutants deserve no interest?

We did not find mutants that are born at a size larger than wild type, divided at a normal size and had a short (birth-size normalized) G1.

Rather, we identified two classes of large-born mutants. The first class, which included many large born mutants, showed a normal (birth-size normalized) G1 duration; namely, mutants in this class spend the same time in G1 as wild-type cells born at the same size. Their average G1 duration was still shorter than normal **average** G1, reflecting their larger birth size. In most cases, compensation was partial and they still budded at an average size that was larger than wild-type cells. In three cases, cells grew slowly in G1 and therefore budded at a size that was similar to wild type (*apl5*, *rpl8a* and *bud21*). These mutants are interesting for studies of regulation of other phases of cell cycle, but in terms of our analysis they do not impact on the size control in G1.

The second class included mutants that were born large and showed a perturbed (birth-size normalized) G1 duration. This class was assigned a role in size control and included e.g mutants in the mitotic exit checkpoint

It appears that they are normalizing for birth size when they determine duration of G1 in the mutant categorization. Is that true? If so, why? Small birth size is an important physiological response (e.g., in poor nutrients birth size -more so than critical size- is reduced, accounting for the longer G1 and delay in initiation of cell division). Normalizing the G1 of various mutants against their birth size (if indeed this is what the authors are doing) introduces a significant bias.

We normalize for birth size when assessing size control because G1 duration depends on birth size, and that this main control point is independent of cell growth rate. We agree that birth size is an important physiological response and indeed it was studied and characterized by others, e.g. in

(Truong *et al*, 2013) which we now cite. However, in the present study we focus on the size control mechanism: namely the compensation mechanism that functions during G1 to compensate for fluctuations in birth size between cells in the population: prolonging the duration of G1 when cells are born small, and shortening the duration of G1 when cells are born large. We are therefore interested specifically in mutants that perturb this control and therefore need to normalize for birth size.

4. The data in Fig 6 about the completely different effects in rpl vs rps mutants on cell size are very problematic. There is no evidence from previous studies that rps mutants as a group have increased size. The Jorgensen et al and the Zhang et al studies in 2002, relying on channelyzer data, show no such trend. If anything, the case is the opposite from what the authors state i.e., rps as a group have smaller overall size than wild type, although there are differences with rpls and within individual rps mutants. In my opinion, the conclusions presented are either a consequence of sample bias in their cherry-picking of mutants, or systematic errors in their assay.

The main point of this figure is the large difference in size and size-control properties between the deletions of RPS and RPL. The average size of the RPS is about the same as that of wild-type, with a relatively small (although statistically significant) increase, as we show. In contrast, RPL deletions are smaller. We emphasize it more in the text.

It is true that Jorgensen and data did not show increase in size in the Rps deletions – (Jorgensen data show p-value < 0.001 that large subunit deletions are smaller and statistically insignificant difference between the deletions in small subunits and the wild type). However, data from microscopy based phenotyping (Ohya *et al*, 2005) show that deletions in large subunits are significantly smaller than the average (p-value < 10^{-4}) while deletions in small subunits are significantly larger than the average (p-value < 0.001) consistent with our results. We now show the corresponding box plots in Figure E7A-C. Other recent study that also measured size using Coulter counter (Moretto *et al*, 2013) observed the same effect that we do (that deletions of small subunits are larger than the wild type).

The study by Zhang et al measured size in stationary cultures therefore the difference between their results and our results taken for growing cells (as well as the results reported above) may reflect differences in growth conditions.

The main message of this figure is that population size (e.g. average cell size) *per-se* is not sufficient for drawing conclusions about size control. By just considering cell size, one would conclude that the large subunits have a stronger role on the cell size control than the small subunits, as their size is significantly smaller. However, the single cell data (correlations between G1 duration and birth sizes) indicates the opposite: small subunits are positive regulators of size control (their deletion significantly extends G1, relative to their birth size), while the large subunits do not play a direct mechanistic role in the size control (affect cell size only through effect on the growth rate).

Our data includes about 35% of ribosomal genes: 18 rpl and 21 rps deletions that were selected through the pre-screen or implicated by other datasets. We are therefore confident in this difference between the large and small subunits.

Further, we corroborate those findings with the phenotype of translation elongation vs. translation initiation factors described by others, which show the respective phenotypes of the rpl and rps genes, respectively, as we describe.

5. Their evidence for a bypass mechanism that controls cell size in the budded phase is rather weak. They report a correlation of 0.24 (again, the type of statistical analysis they used is not clear), but this is hardly a strong support for the mechanism they propose. If I am not mistaken, the Di Talia et al 2007 study that used similar methodology, which they cite, found no evidence for size control in the budded phase. Finally, the whi3 and whi5 mutant analyses they show are indirect and they do not directly address the question whether there is a cryptic size control in the budded phase.

di Talia et al did not study size control in the budded phase and do not relate to that. We agree that in the wild type, the size control at the budded phase is weak, although clearly significant. This is

probably since cells are rather large, due to the size control that functions during G1. This weakening of size control in large cells is analogous to the observation by di Talia and colleagues showing that the size control in G1 phase is weak when cells are large.

The main observation that makes us confident that size control in the budded phase is relevant, is the phenotype mutants that bud at a small size (e.g. *whi5* and *whi3* but also others): here, the size control becomes much stronger and is in fact equivalent in strength to the size control that functions in G1. We interpret this increased significance by the fact that cells are smaller and more diverse in sizes when entering the budded phase, hence size control is now needed.

Note that the figures were improved based on reviewers requests and our realization that the previous version was less straightforward to interpret.

-Specify experiments or analyses required to demonstrate the conclusions

1. Overall, the relevant sections of their pre-screen as it is done and presented has very little value. In addition to their raw flow cytometry data, they should collate in a separate dataset all the relevant values from all these studies and theirs, side by side, and present actual numerical analyses of their correlations, so the readers can easily compare these studies and properly evaluate the authors' conclusions. After such an analysis, it is possible (perhaps even likely) that their arguments for using FSC as a pre-screen for cell size will not hold much water. In that scenario, the data will still be valuable to the field, as a side-by-side comparison of these different approaches to query "cell size". If that is the case, they should reformat this section not as an accurate pre-screen for their microscopic analysis that follows, but as a comparative analysis of methods that report on size, define how they selected their 591 mutants and move on to the more interesting parts of the manuscript.

As suggested, we reduced the focus on the prescreen and now stress more the point that all methods of size estimation in population have their drawbacks and we chose the 591 mutants combining information from different screens.

We write in the results section

We verified the reproducibility of our measurements by repeating the analysis for 750 small and 750 large strains (Figure 1C) and compared our results to previous screens (Figure 1D and Dataset E1). 23 of the 26 strains previously assigned the *whi* (small size) phenotype, had average size below median ($P < 10^{-5}$) and one (*ygr064w*) didn't grow well (Figure 1D-E). Overall, correlations between results of different screens were significant, but relatively low, stressing the difficulty of measuring cell size in high throughput manner and the strong effect of environmental conditions on the average cell size.

And

We selected strains with small size and relatively short G1 as candidates for being negative regulators and strains with a large size and relatively long G1 as candidates for being positive regulators (Figure 1F and Expanded View Text Section 4). To overcome noise in size measurements, we used size estimations either from our pre-screen and its repeats or electronic volume measurement data from the screen by (Jorgensen *et al*, 2002). This way, we defined 255 putative negative and 264 putative positive regulators. We supplemented this list by strains involved in the ribosomal biogenesis and additional strains previously implicated in the regulation of START. Overall, a list of 591 candidate strains was assembled (Dataset E2).

As requested, we have added to the Dataset E2 results of sizes and G1 percentages measured in other screens. We apologize for the omission of the actual correlations from the previous version of Figure 1. The correlation values were now added.

All the flow cytometry data were uploaded to the FlowRepository, a public repository for flow cytometry data. The data is currently open to the editor and the referees, and will be available to everyone upon publication. Please use the following access keys for the data (the whole pre-screen was split into ten chunks because of the database requirements):

Dataset	Link
pre-screen 1/11	http://flowrepository.org/id/RvFrhTFfc2lQ8Red1LbnLUuP3xQNr8Jp9j90POTUNDyqO02mdQ83YpUEx7UroFa6
pre-screen 2/11	http://flowrepository.org/id/RvFrKt7Foi536vAfzAwsYxPcLf41NH66bLJ2Llc15Gvzwuo3eWK8baXLxj7GbouC
pre-screen 3/11	http://flowrepository.org/id/RvFrk75qGZU446lLNFFVJycEoWsgCGdDz7EGGnddKtksE0NVSmCmU45duOAcwpc3
pre-screen 4/11	http://flowrepository.org/id/RvFr4tb3psbHfBn1twVFSI0tAZ46BaHHQ95GtlMaOJLdj2Fiwo1iiBsiOqxMT9R7
pre-screen 5/11	http://flowrepository.org/id/RvFr8MJMZ1zZjrii0ykrxQ2InOLPGQoZIFW1xEjwOU8ZzMa1w5xZ6Bge5Cf2OwiA
pre-screen 6/11	http://flowrepository.org/id/RvFr8eUPbrafrBW19J3fe9ocL2FYlCpnphPIUHeF9CUp6oTYhlQcIjj7CSWqX9Cr
pre-screen 7/11	http://flowrepository.org/id/RvFrusAeuqIRBSrB216eklqfcM3xcHKmfBKIEEvQIMxwsfjAucNFqplnD3rOOk3b
pre-screen 8/11	http://flowrepository.org/id/RvFr3kb8Ikjam2abCKb618dJPSH2nPDW7fy9tYumUAVT4FfT47GCV5bH4Jg1phh2
pre-screen 9/11	http://flowrepository.org/id/RvFrikP3dPzAdisqrXiBl2Mnt0W39ffQb8iXncv0g5911C9J7ws3fTNEngadR9sq
pre-screen 10/11	http://flowrepository.org/id/RvFrVY49rJZOrGXqjG7nP0b4y5mxQYoBwjleRkrPpCKMa8PUeC32KjJNMZei5IHu
pre-screen 11/11	http://flowrepository.org/id/RvFrUvz2z2xCmzfBIBi7GPEHwdvC0kCLLZuYtHmnTQ5100UE1DI0Fn2zaQZ7JjOH
Repeat extended G2	http://flowrepository.org/id/RvFredj5gp4vGmJsHGOxPCKKYfr0CZUCJhWScI0o8JvQXr901GUpu2DgrhOsHceS
Repeat extended G1	http://flowrepository.org/id/RvFr0wgMeq07R1A4pIEG5smznYLGyG1TCK6RDlSAprnfwLEMjwIUpw6sziyCeGzt
Repeat extended S	http://flowrepository.org/id/RvFrBwv03baWU9RYGYIs6KjOggySz2QwAQttpDT9iB1EPBeilZKPOW2iUDPvi5Kc
Repeat small size	http://flowrepository.org/id/RvFr0vXV5ihXKKcs3YsIdQ9fJf33rcHuuxmmfEDNuJ2JpH1wrxMIQj1JBdInLxtx
Repeat large size	http://flowrepository.org/id/RvFrJTg7z6qNG5AZsY71CvgTzutXHbMDvffP5gULuFbyZlIwsbyhTQr5UYsZwUAI
Repeat special interest	http://flowrepository.org/id/RvFrTVbuDQ7pdrDUiHNGs3pRUbmulu1UgZpSZiLtSE02i575Dfw6C0rQmO0XR92C

We extended the description of selection of candidate strains in the Extended View Text:

For the time-lapse microscopy screen we defined six different (partially overlapping) lists (see below, and Expanded View Dataset E2), introduced fluorescent reporters by SGA methodology and measured cell cycle distribution of the resulting strains. If the cell cycle distribution was significantly different from the original strains (e.g. only diploid colonies recovered after the SGA, or no progeny recovered), we repeated the SGA.

Negative regulators based on flow cytometry size measurement

We plotted median forward scatter versus percentage of cells in G1 for each mutant in the main screen. This produced a negative correlation as shown on Figure 1F. We neutralized this negative correlation and selected all strains that had forward scatter below expected from the negative correlation between the G1 percentage and forward scatter. Then we selected these strains and measured their cell size and cell cycle distribution. In this repeat every plate contained 4 controls of the wild type strain. Only strains that had the median forward scatter lower than the lowest repeat of the wild type strain were taken for the microscopic screen. Candidate strains were ordered by increasing G1 percentage and for the microscopic screen we took all strains with percentage of G1 phase below that of the wild type (150 strains). 12 strains did not maintain their phenotype after SGA.

Positive regulators based on flow cytometry size measurement

We plotted median forward scatter versus percentage of cells in G1 for each mutant in the main screen. This produced a negative correlation as shown on Figure 1F. We neutralized this negative correlation and selected all strains that had forward scatter above expected from the negative correlation between the G1 percentage and forward scatter. Then we selected these strains and measured their cell size and cell cycle distribution. In this repeat every plate contained 4 controls of the wild type strain. Only strains that had the median forward scatter above the highest repeat of the wild type strain were taken for the microscopic screen. Candidate strains were ordered by decreasing G1 percentage and for the microscopic screen we took all strains with percentage of G1 phase at least 15% above that of the wild type (194 strains). 20 strains failed to maintain their size after the SGA

Negative regulators based on Coulter counter size measurement

We plotted median cell size from (Jorgensen *et al*, 2002) versus percentage of cells in G1 for each mutant in the main screen. This also produced a negative correlation. We neutralized this negative correlation and selected all strains that had cell size below the expected from the negative correlation between the G1 percentage and median cell size. Candidate strains were ordered by increasing G1 percentage and for the microscopic screen we took all strains with percentage of G1 phase below that of the wild type (136 strains). 17 strains did not maintain their phenotype after SGA.

Positive regulators based on Coulter counter size measurement

We plotted median cell size from (Jorgensen *et al*, 2002) versus percentage of cells in G1 for each mutant in the main screen. This also produced a negative correlation. We neutralized this negative correlation and selected all strains that had cell size above the expected from the negative correlation between the G1 percentage and median cell size. Strains that had the median forward scatter lower than the lowest repeat of the wild type strain were taken for the microscopic screen. Candidate strains were ordered by increasing G1 percentage and for the microscopic screen we took all strains with percentage of G1 phase below that of the wild type (115 strains). 30 strains did not maintain their phenotype after SGA.

Known regulators of cell size and cell cycle

All *whi* strains from (Jorgensen *et al*, 2002), *whi* and *lge* strains from (Zhang *et al*, 2002) and all strains deleted of genes that have GO annotation "regulation of cell size". (103 strains, 19 failed SGA)

Genes implicated in ribosomal biogenesis

Deletions of all non-essential genes that were annotated with GO term 'ribosomal biogenesis' or its descendants as of May 7, 2009 (117 strains, 7 did not undergo SGA).

2. Present independent and separate analyses of volume growth rate as a linear and as an exponential function, both for wild type cells in different nutrients and for their mutants, and draw appropriate conclusions. Also, they need to cite and correlate their study with others that reported on similar topics. Especially with their microscopic analysis, they need to refer to Kang et al (Integr. Biol., 2014, DOI: 10.1039/C4IB00054D). For their birth size measurements they need to correlate with the values reported in Truong et al (G3. 2013 Sep 4;3(9):1525-30).

We added an analysis of the growth rate (assuming either linear or exponential growth) to the SI (Figure E3A-B). Our conclusion from this analysis is that cells are growing exponentially. This is because when we predicted growth rate while assuming linear growth, we obtained correlation between the birth size and the predicted growth rate. In contrast, when predicting the specific growth rate assuming exponential growth, no such correlation was identified.

We mention this result of exponential growth in the text when first mentioning cell growth rate. The suggested references are now cited.

3. The categorization of the mutants shown in Fig. 4 needs improvement. They need to give specific examples of mutants, explain the statistics, and better explain why they are classified as important or not for their conclusions.

As explained above, we significantly expanded this section and added Figure E4 that clarifies statistics that we have done.

4. To make the claims they make in Fig. 6 about large vs. small rp mutants, they need to include all the mutants, analyze them with repetitions and perform very robust statistics. As it stands, the reported larger than normal mean size and budding size for rps mutants is unsubstantiated and contradictory to previous independent studies. The onus is on the authors to convince the field otherwise.

As mentioned above, our analysis included 35% of ribosomal genes: 18 rpl and 21 rps deletions. In the previous version of figure 6, we provided examples of specific deletions to stress the effect. We now added panels to the same figures that show relative change in the length of G1 and in volume increase in G1 for all ribosomal mutants examined (Figure 6 E,F) together with the density plots focusing on the slow growing mutants. The difference is also stated in the figure caption.

We further repeated our measurements of all ribosomal strains that show a significantly perturbed size at budding, 15 repeats of deletions in small subunits and 14 repeats of deletions in the large subunit. The results are now shown on Figure E7C. We compared the birth sizes, budding sizes, birth-size normalized G1 lengths and birth-size normalized DVs. The Pearson correlation coefficients were 0.55, 0.66, 0.7 and 0.65 respectively, similar to other mutants. We have performed various statistical tests examining if the differences between the deletions in the small and the large subunits are significant and the results are shown below (Also added as Table E4).

Test	Statistic	Result	p-value
Size at budding similar between RPL and RPS	t-test	Reject H_0	$1.6 \cdot 10^{-5}$
Size at budding similar in RPL and in wt	t-test	Reject H_0	10^{-5}
Size at budding similar in RPS and in wt	t-test	Reject H_0	10^{-11}
Normalized G1 similar between RPL and RPS	t-test	Reject H_0	0.007
Normalized DV in G1 similar between RPS and wt	t-test	Reject H_0	$2.5 \cdot 10^{-4}$

5. In the absence of any new data, they would need to modify extensively their arguments about cryptic size control on the budded phase, in various places in the manuscript, including in the abstract. As it is now, they try to make too much out of a very weak result (a correlation of 0.24).

As we noted above, size control in the budded phase is indeed weak in wild type cells (likely because of their already large size) but becomes significantly stronger in small mutants, similarly to the behavior reported by DiTalia for the well-studied size control in G1. For example, in the small mutants *whi5* and *whi3* the correlation between the size at budding and the length of the budded phase is -0.35 and the correlation is -0.42 when all small mutants are united. In fact, in those mutants, the strength of this size control at the budded phase is close to that of the size control in G1. We improved the figures and the writing to demonstrate this result better.

Minor points

-Easily addressable points

1. Please refer to the "Dataset" files as datasets, not as Supplementary tables, so it is easier to follow. Consistency in the labeling (in whichever way the authors prefer) in the text helps the reader.

Corrected

2. Column C in Dataset 1 has no label (should be Normalized G1?)

Corrected

3. In the subsection "Effect of translation and ribosomal biogenesis...", the first paragraph needs specific citations to back up their statements.

Corrected

4. In the next paragraph from the one mentioned in the previous minor comment, they start with the statement: "The interpretation of these results was complicated by a general lack of single cell data making it hard to distinguish the direct from the indirect effects". Why is that so? I do not think that the controversy regarding the interpretation of ribosome mutant phenotypes has anything to do with single-cell vs. population-based data.

We hope that we were able to clarify why single cell data is useful for assigning function for mutants affecting cell size.

-Presentation and style

Style is fine

-Trivial mistakes

Check the text a little more carefully. When they reference something, it should be there. Also, articles (a/the) seem to be missing at several places.

For major revision, it is useful if you can provide a time estimation for the requested additional experiments/analyses.

I do not know. Depending on the throughput of their assay, 2-3 months might be enough.

Reviewer #2:

In this manuscript, the authors report the results of a high throughput screen for cell size mutants. Unlike previous such screens, the authors examine single cell correlations between the size of cells and budding and their size at birth as determined by automated segmentation of time-lapse movies. This allows the clean separation of size mutants that are small because they are giving birth to very small daughters rather than affecting the size control mechanism gating the G1/S transition. Thus, the screen is a clear improvement upon previous such efforts. The authors were able to uncover an interesting piece of biology in that mutations affecting the small subunit of the ribosome and translation initiation factors had a clear effect on G1/S control, while mutations in the large subunit, while they did affect population size and growth, did not affect G1/S control. This result might prove important in the determination of the molecular mechanism through which cell size is transduced to gate the G1/S transition, an important and ill-understood piece of biology. Following some minor revisions suggested below, this work should be published in MSB.

We thank the reviewer for the positive evaluation of our work and for useful suggestions that improved the paper.

Figure 2: I found the density plots, especially when overlaid with 4 curves, quite difficult to use to compare different experiments and mutants. I think the presentation might be more clear if the authors plot the data using box plots after binning the data. Also, in this section the authors grow cells on 4 different environments, but none of them are really large perturbations in growth rate. If the authors want to claim support of a timer-type model, it would perhaps be more useful to examine cells growing much more slowly, such as on ethanol.

We made contour plots on Figures 2, 6 and 7 weaker, and stressed the lines that we now show with standard deviations of median per bin to facilitate comparison between mutants and conditions. We prefer showing the contour plots so that one can easily estimate the strength of the effect relative to the natural variability.

The main reason to compare different environments was to be able to compare between mutants that grow at different proliferation rates. Since almost all our mutants grow with doubling times less than

130 min (Figure E3F), these proliferation rates are covered by the conditions examined. At slower growth rates (ethanol, data not shown) we do see the shift of the dependency of length of G1 on the birth size upwards, and we are not sure if the reason is in different physiology of cells growing in respiratory conditions or the change in parameters of the size control at slower growth rates. (note that for this reason we classified the outlying ten strains whose growth rate falls outside of the growth rates of the four media shown [extended relative G1, decreased relative volume increase in G1] as ambiguous).

Figure 4: It would be better to plot Size in fl, which should be the same for everyone, than pixels, which are not.

We agree and change the units.

Figure 5: Have the authors examined the recently published whi7 mutant from the Aldea lab (Mol. Cell)? It would be great to see how that mutant affects size control in the authors single-cell assay. Also, the y-axis of many panels in C have been clipped during some cutting and pasting to make the figure.

Deletion of *WHI7* was examined in the pre-screen, but did not pass to the microscopy screen because it did not pass the cutoffs in flow cytometry measurements and similarly appeared to be of normal size also in the other published screens. We now cite this paper in the Introduction. The figure was corrected.

Figure 7: The WT data density plot is shown 3 times, which seems excessive especially. I have the same comments as for figure 2, where I think a more standard box-plot after binning by budding size would allow an easier comparison between mutants budding at the same size.

We agree and changed the figures.

Last paragraph in the section titled 'Microscopic screen of...' refers to Fig 2F and G, when it should be Fig 3C, D.

Corrected

First paragraph of the section 'Effect of translation and ribosomal...' is missing all references for the work described.

Corrected. We now cite the following papers: (Popolo *et al*, 1982; Moore, 1988; Hartwell & Unger, 1977; Jorgensen *et al*, 2002, 2004; Moretto *et al*, 2013)

Reviewer #3:

The manuscript, "Systematic Identification of regulators of size in budding yeast using high throughput time-lapse microscopy" by Soifer and Barkai, describes a microscopic screen of the genome-wide deletion collection for mutants affecting cell size, largely during G1 phase, and their implications in terms of understanding cell size regulation. Importantly, the data are evaluated in terms of the relationship between size at birth, size at budding, growth during G1 and time in G1. That comparison showed that whereas the growth during G1 was dependent upon both the birth size and the growth rate, the length of G1 was only dependent upon birth size. Interestingly, this can account for the difference in budding size observed in different carbon sources. Comparing the effect of mutants on the relationship between G1 length and birth size, led to the classification of genes based upon their negative and positive effect on the size at bud emergence. Among those genes in which mutation causes an advancement in budding were several known to encode negative regulators of Start along with several for which such a role had not been previously described. Similarly, mutations in genes encoding known positive regulators of Start were observed to delay budding both temporally and in terms of growth during G1. However, when ribosomal protein genes were classified based upon their effect on Start, surprisingly, mutations in those encoding large subunit constituents advanced Start, whereas mutations in those encoding small subunit constituents

delayed it. The authors conclude that elements of the large subunit appear to act as negative regulators of Start because diminished protein elongation limits cell growth during G1 and, therefore, leads to bud initiation at a small cell size, whereas those encoding elements of the small subunit appear as positive regulators because the cell cycle is positively regulated by translation initiation. Finally, the study shows that when cells pass Start at a small cell size, an additional size control mechanism is revealed late in the cell cycle.

This is a carefully executed single cell analysis of cell size control that attempts to shed some of the constraints of previous screens to discover new size control elements and uncover new modes of regulation of cell size. Furthermore, this study preselected a relatively larger set of mutants for analysis by single cell microscopy than had prior studies using an improved flow cytometric approach. That said, the novelty of the findings is relatively limited. The screen has revealed several genes not previously recognized to play a role in determination of size and uncovered an apparent bifurcation between the effects of mutation of large and small ribosomal protein genes on cell size at budding. Unfortunately, they have not further characterized the roles of these genes so it remains unclear whether they are direct regulators of cell size or affect the process indirectly. In addition, the authors neglect to tightly relate their work to the relatively extensive literature on cell size regulation based upon both population and single cell analysis. For example, they only fleetingly refer to models that distinguish between "timers" and "sizers", well accepted mechanistic models for size control, and otherwise discard commonly understood terminology, such as Start and bud emergence in lieu of the more ambiguous G1/S. Furthermore, although the behavior of known size control mutants appears to conform expectations, it is unclear precisely how their analysis relates to that in recent studies. This is important because previous studies seem to have established the independence of cell size determination from the birth size, whereas this study argues that birth size is a major determinant of size at budding. Finally, there are numerous unclear or poorly constructed arguments and many grammatical errors throughout the text, especially in the discussion. These issues are detailed below, along with a number of other issues, and will need to be addressed prior to publication of this work.

We thank the reviewer for the overall positive evaluation of our work and for the helpful comments. We answer the critical comments below.

Specific points:

1. These authors express surprise that the effect of cell size at birth on the length of G1 (time from birth to budding) is independent of growth rate. However, Ferrezuelo et al, 2012 (Figure 3) show little correlation between the size at birth and the time spent in pre-start G1 phase or of the effect of carbon source on that relationship. Instead, they and di Talia et al, 2007 show a strong correlation between growth rate and size at Start. It is unclear whether this is a discrepancy between the results or, rather, one of interpretation. Although there is a difference between the studies in the cellular events measured (see below), it seems doubtful that this is sufficient to explain any discrepancy.

A. Ferrezuelo et. al did not report the correlation between size at birth and the time spend in G1. In their figure 3 they focus the added volume (growth rate * G1_duration) in G1. They do show that there is no correlation between G1 duration and growth rate, and also that size at Start correlates with growth rate (Fig. 2). Both those findings are completely consistent with what we see. One of our key finding is that the time until start depends on birth size but does not depend on growth rate. Consequently, growth rate effects the size at START (since faster growing cells add more volume at the same time) This can be seen, for example, when looking at cells born at a similar size, cells that grow faster bud at a larger size (Figure E3C). This is also shown on the main text figures 2D,E: since lines on the plot of birth size vs. volume growth in G1 shift upwards with increasing growth rates, cells that are born at the same size grow more in volume during G1 when growth rate is larger, and consequently bud larger.

B. In agreement with our results, di Talia et al show that birth size is a major determinant of volume increase in G1. In fact, they see that 32% (haploids) or 45% (diploids) of the variability in the duration of G1 phase can be attributed to the effect of the birth size. In our data the correlation

coefficient between the size at birth and the volume growth in G1 is -0.58 in haploid and -0.61 in diploid cells, meaning that the size at birth explains 33% (haploids) or 36% (diploids) of the variability in the duration of G1. We added a note to the supplementary material comparing our data to the other datasets (Table E1).

C. finally, we note that neither study examined the connection between birth size and G1 duration, which we find to be invariant to growth rate.

2. The authors have chosen to measure septin ring dissolution and assembly as indicators of G1 entry and exit. Other recent studies have chosen to measure Whi5 exit from the nucleus as the first indication of cell cycle commitment or Start. The authors should discuss their results in the context of those differences. For example, is it possible that the cell size phenotype of some mutants a consequence of defects in the timing of septin ring formation rather than execution of Start?

Thank you for this comment. We mention this point in the Results section:

However, since properties of size control are similar if measured at budding or e.g. via Whi5 localization (Di Talia *et al.*, 2007), we decided to use bud emergence as a reporter to START, since its strong signal enabled automated detection and analysis. In a typical experiment, we followed sixty fields of view (up to 12 different strains) (Figure 2A) and used an image analysis software that we developed to automatically track individual cells, identify the cell cycle transitions and build cell lineages (Figure 2B-C and Figure E2).

We also mention in the discussion that some of the mutants could regulate G1 length downstream of START.

Note that since we measured G1 length and not the execution of START, some of the mutants could affect budding and not START. Most of the identified regulators, however, do not belong to functional categories that seem likely to decouple those two processes.

In addition, the difference in the phenotype measured raises a problem with terminology. This study does not formally measure cell size at Start but it also does not measure cell size at G1/S, the term used by the authors. S phase may coincide with septin ring formation under some growth conditions, but certainly not under all or in all mutants. It is probably best to refer to this as bud emergence.

We now use term Start instead of G1/S transition that is the primary size control in the introduction and discussion. In the results section we use the term “budding”. The reason we chose to measure the septin ring and not e.g. localization of Whi5 is that the septin ring is much more amenable to automatic analysis. We now explain more precisely that we measure budding and not START.

3. The authors should compare and contrast their model with current models for size determination. Do these findings indicate that the cell size at budding is primarily determined by a "sizer" or "timer" mechanism or is instead explained by some other mechanism.

We now emphasize the connection of our work to the classical models. In our view, the distinction between timers and sizers is not binary. The limiting cases are when the duration of the phase is set only by the current cell size (“sizer” or checkpoint) or completely independently of cell size (“timer”). However, our results suggest that size control can be achieved by a spectrum of mechanisms. Both at budding and at the budded phase we observe incomplete compensation of size fluctuations: size at budding (or at the end of budded phase) correlates with the size at birth. This means that not only size controls the duration of the phase but also other variables such as initial size, time since the start of the phase etc. In particular, in the budded phase we observe transition from size independent phase (“timer”) to partially size dependent phase (mixture of “timer” and “sizer”). This “mixture” model or “weak” size control had been previously proposed theoretically, we briefly analyze this model and show that it ensures size homeostasis. We tried to do so in the end of the discussion of the size control in the budded phase, but failed to do it clearly. We now say:

Strengthening of size control in small cells questions the classical distinction between "timers" and "sizers". Timers are phases of the cell cycle that do not depend on cell

size, while "sizers" are phases that are size dependent. Our results suggest that this distinction is arbitrary. It seems that all phases of the cell cycle could be timers or sizers depending on cell size. Perhaps when cell size is small, some cellular components are becoming limiting for cell cycle progression making the length of this phase size-dependent. In large cells, the same phase becomes a timer since these components are no longer limiting. In this model, cell size affects the rate of cell cycle progression, instead of being a requirement for transitions between phases, similar to models proposed mathematically (Chen *et al*, 2000; Pfeuty & Kaneko, 2007; Charvin *et al*, 2009). As previously argued, this alternative mode of size control is sufficient to ensure size homeostasis under conditions of exponential growth (Tyson & Hannsgen, 1985; Csikasz-Nagy *et al*, 2006). In the Expanded View Text section 5 we briefly analyze this model of size control and show that it ensures size homeostasis.

4. There is a relatively extensive literature concerning the effect of mutants in ribosomal protein genes and ribosome biogenesis genes as positive and negative regulators of Start. There are mutants in translation initiation factors that would seem to support a positive role for translation initiation in the regulation of Start (cdc68, for example). The authors should discuss whether studies or the effects of inhibitors of translation initiation and elongation are consistent with their observations and, perhaps, test the effect of such inhibitors on cell size in the context of their experiments.

We added a paragraph to the discussing role for translation in general and translation initiation in particular, on START to the introduction. We also discuss the connection of these findings to our results, in both the results section and in the discussion. In agreement with the positive role for the translation initiation in START we observe that many deletions of initiation factors increase cell size, suggesting that they translation initiation but not elongation plays a positive role in START (Figure 6H).

5. Discussion: The section "Role of protein synthesis..." is quite confusing. First, the two models for the role of protein synthesis in the regulation of cell size presented by the authors both posit a positive role for protein synthesis but the second model has an additional condition, an opposing role for ribosome biogenesis. The same paragraph then goes on to provide two pieces of evidence in support of "this" model. Which model are they referring to? Presumably the second model. That should be clarified. Second, the next paragraph goes on to argue that their data supports the first model, despite the fact that they have just provided their own data in support of the second model. Although I understand the arguments that are made, the construction of these two paragraphs makes the assignment of the various arguments to different models ambiguous.

We completely rewrote this section. We agree that it was confusing in its previous form, we do hope that it is now written more clearly. Here is the new text:

Protein synthesis had long been implicated as a positive regulator of START (Popolo *et al*, 1982; Moore, 1988). Inhibition of protein synthesis delays START, causing cells to bud at a larger size. Although the details of how protein synthesis promotes START are not completely clear, at least partly it acts through Cln3. An upstream ORF in the Cln3 mRNA inhibits its translation (Polymenis & Schmidt, 1997). Due to this upstream ORF, the translation of Cln3 is affected disproportionately relative to other proteins when the overall protein synthesis capacity is reduced.

As had been pointed out, this model is not without certain difficulties (Turner *et al*, 2012; Jorgensen & Tyers, 2004). If the overall translation rate stimulates START, one would expect that in poor growth conditions or when ribosomal content is decreased, cells would also delay START and increase their size. In general, however, the opposite is observed: poor nutrient conditions (Johnston *et al*, 1979) or deletions affecting the ribosome (Yu *et al*, 2006; Jorgensen *et al*, 2002, 2004) decreased the average cell size. This led to the suggestion that while translation itself is a positive regulator of START, the rate of the ribosomal biogenesis has a negative role in START (Jorgensen & Tyers, 2004). In particular, since deletions of proteins involved in the assembly of the large ribosomal subunit (structural proteins or biogenesis factors) decrease cell size to a larger extent than factors of the small ribosomal

subunit, it was proposed that START depends on the flux through the pathway producing the large subunits (Dez & Tollervey, 2004; Moretto *et al*, 2013).

Our results suggest a unified explanation for those findings. We observed that deleting components of the large ribosomal subunit or of genes involved in ribosomal biogenesis does not affect the actual size control. Rather, cells become smaller simply because they grow slower. In contrast, parts of the small ribosomal subunit behave as positive regulators of START, extending G1 duration more than expected given their birth size, and consequently budding at a size comparable or larger than wild type cells born at a small size. This likely reflects their distinct role in translation initiation, not shared by the large ribosomal subunit. Our results therefore suggest that translation initiation is a positive regulator of START, hence its inhibition, as observed in the initial experiments, prolong G1 and could lead to a larger budding size. In contrast, translation elongation affects predominantly the cell growth rate, and therefore decreases cell size, as observed upon deletion of ribosomal components. We note that this role of translation initiation in promoting the START transition is supported by multiple studies: deletion of eIF4 (cdc33), eIF3 (cdc63) prolongs G1 and increases cell size (Brenner *et al*, 1988; Hanic-Joyce *et al*, 1987; Polymenis & Schmidt, 1997) and many strains depleted of translation initiation factors have an increased size (Figure 6H)

6. The existence of a second size control point during G2/M that is revealed when the budding size is small has been previously reported.

Thank you for noting that. We weren't aware of those reports, but now mention them in the discussion:

Previous evidence suggested that G2/M morphogenesis checkpoint can also act as a cryptic size control activated e.g. when the bud is not large enough (King *et al*, 2013; Anastasia *et al*, 2012; Rupes, 2002; Harvey & Kellogg, 2003). We do not know if the phenomenon we observe is related to morphogenesis checkpoint. Our results are reminiscent of the cryptic size control point identified in fission yeast in small *wee1* mutants (Fantes & Nurse, 1978). Note that in the fission yeast, both the primary and the cryptic size control comply with the checkpoint paradigm (Sveiczer *et al*, 1996), while in budding yeast both size controls compensate only partially for size fluctuations.

Minor points:

We corrected errors and added missing references. The paper underwent significant changes and we took extra care to fix grammatical errors and typos. We apologize and thank you for pointing this.

1. Should cite recent paper showing that retention of *Cln3* at the ER is dependent upon *Srl3/Whi7* (Aldea lab).

Citation added

2. Regulation of Start by regulators of ribosome biogenesis (*Sfp1*, etc.) should be mentioned in the introduction.

Corrected

3. Page 3, line 4: Should read "The quantitative...." rather than "Quantitative...."

Corrected

4. Page 6, paragraphs 2 and 3: The figure numbering is incorrect. Figure 2 F and G should be Figure 3 C and D. Figure 4 does not have A and B components.

Corrected

5. Page 7: Should mention other reported functions of *Whi3* (Aldea lab).

Mentioned in the introduction

6. Page 7: Paragraph discussing relevance of the connection between *Rsr1* and *Lte1* is highly speculative and, if it belongs anywhere, it should be in the discussion.

Moved to the discussion

7. Page 8: Numerous studies are mentioned without reference in the first paragraph of the section "Effect of translation...." These references should be cited. Also, the last paragraph should start with "A recent...."

Corrected

8. Page 8: It may be appropriate to cite and discuss Thapa et al, 2013 which catalogs the effects of RP mutations on cell morphology, cell cycle distribution, etc.

The reference was mentioned and cited in the introduction

9. It would be helpful to distinguish between cell "growth" and "proliferation" throughout the manuscript.

We made an effort to distinguish between growth and proliferation. In some cases (when we refer to both growth and proliferation) we use the term growth.

10. Page 9, paragraph 1 of second section: The third sentence is confusing. "...alternatively, some process starts at the beginning of the cell cycle and completes until the cell undergoes division." Please clarify.

The sentence now says:

First, the overall duration of the cell cycle may be controlled e.g. by some process which is initiated at cell birth and has to be completed before cell divides.11. Page 11, paragraph 2: "...five hundreds..." should be "...five hundred..." "Large proportion..." should read "A large proportion..."

Corrected

12. Page 11, paragraph 4: "...longer G1 that..." should read "...longer G1 than..."

Corrected

13. Last sentence of Discussion: "...size homeostasis at the conditions of an exponential growth..." should presumably read "...size homeostasis under conditions of exponential growth."

Corrected

14. Page 12, last paragraph: The 5th sentence refers to a second model but there is no reference to two models of size control earlier in the section. Is this a reference to models in the previous section. If so, it should be made clearer.

Corrected

15. There are many other typographical and grammatical errors throughout. The paper should be carefully edited.

We made many changes to the paper, improving the writing and correcting errors.

Point-to-point references

Anastasia SD, Nguyen DL, Thai V, Meloy M, MacDonough T & Kellogg DR (2012) A link between mitotic entry and membrane growth suggests a novel model for cell size control. *J. Cell Biol.* **197**: 89–104 Available at: <http://jcb.rupress.org/content/197/1/89.full> [Accessed May 25, 2014]

- Brenner C, Nakayama N, Goebel M, Tanaka K, Toh-e A & Matsumoto K (1988) CDC33 encodes mRNA cap-binding protein eIF-4E of *Saccharomyces cerevisiae*. *Mol. Cell. Biol.* **8**: 3556–3559 Available at: http://mcb.asm.org/content/8/8/3556.abstract?ijkey=1fd67c1dcf55fc9dce2338714c717b1237587a38&keytype=tf_ipsecsha [Accessed June 3, 2014]
- Charvin G, Cross FR & Siggia ED (2009) Forced periodic expression of G1 cyclins phase-locks the budding yeast cell cycle. *Proc. Natl. Acad. Sci. U. S. A.* **106**: 6632–6637 Available at: http://www.ncbi.nlm.nih.gov/entrez/query.fcgi?cmd=Retrieve&db=PubMed&dopt=Citation&list_uids=19346485
- Chen KC, Csikasz-Nagy A, Gyorffy B, Val J, Novák B & Tyson JJ (2000) Kinetic Analysis of a Molecular Model of the Budding Yeast Cell Cycle. *Mol. Biol. Cell* **11**: 369–391 Available at: <http://www.molbiolcell.org/cgi/content/abstract/11/1/369>
- Dez C & Tollervey D (2004) Ribosome synthesis meets the cell cycle. *Curr. Opin. Microbiol.* **7**: 631–7 Available at: <http://www.sciencedirect.com/science/article/pii/S1369527404001316> [Accessed June 4, 2014]
- Fantes PA & Nurse P (1978) Control of the timing of cell division in fission yeast. Cell size mutants reveal a second control pathway. *Exp. Cell Res.* **115**: 317–329
- Fortuna M, Sousa MJ, Côrte-Real M, Leão C, Salvador A & Sansonetty F (2001) Cell cycle analysis of yeasts. *Curr. Protoc. Cytom.* **Chapter 11**: Unit 11.13 Available at: <http://www.ncbi.nlm.nih.gov/pubmed/18770687> [Accessed June 4, 2014]
- Haase SB & Reed SI (2002) Improved flow cytometric analysis of the budding yeast cell cycle. *Cell cycle* **1**: 132–6 Available at: <http://www.ncbi.nlm.nih.gov/pubmed/12429922> [Accessed January 25, 2012]
- Hanic-Joyce PJ, Singer RA & Johnston GC (1987) Molecular characterization of the yeast PRT1 gene in which mutations affect translation initiation and regulation of cell proliferation. *J. Biol. Chem.* **262**: 2845–51 Available at: <http://www.ncbi.nlm.nih.gov/pubmed/3029094> [Accessed June 4, 2014]
- Hartwell LH & Unger MW (1977) Unequal division in *Saccharomyces cerevisiae* and its implications for the control of cell division. *J. Cell Biol.* **75**: 422–435 Available at: <http://www.jcb.org/cgi/content/abstract/75/2/422>
- Harvey SL & Kellogg DR (2003) Conservation of mechanisms controlling entry into mitosis: budding yeast *wee1* delays entry into mitosis and is required for cell size control. *Curr. Biol.* **13**: 264–75 Available at: <http://www.ncbi.nlm.nih.gov/pubmed/12593792> [Accessed July 30, 2014]
- Johnston GC, Ehrhardt CW, Lorincz A & Carter BLA (1979) Regulation of cell size in the yeast *Saccharomyces cerevisiae*. *J. Bacteriol.* **137**: 1–5 Available at: http://www.ncbi.nlm.nih.gov/entrez/query.fcgi?cmd=Retrieve&db=PubMed&dopt=Citation&list_uids=368010
- Jorgensen P, Nishikawa JL, Breikreutz B-J & Tyers M (2002) Systematic identification of pathways that couple cell growth and division in yeast. *Science (80-.)*. **297**: 395–400 Available at: <http://www.ncbi.nlm.nih.gov/pubmed/12089449> [Accessed July 10, 2011]
- Jorgensen P, Rupes I, Sharom JR, Schnepfer L, Broach JR & Tyers M (2004) A dynamic transcriptional network communicates growth potential to ribosome synthesis and critical cell size. *Genes Dev.* **18**: 2491–505 Available at:

<http://www.pubmedcentral.nih.gov/articlerender.fcgi?artid=529537&tool=pmcentrez&renderType=abstract> [Accessed July 18, 2011]

Jorgensen P & Tyers M (2004) How cells coordinate growth and division. *Curr. Biol.* **14**: R1014–R1027 Available at: <http://www.sciencedirect.com/science/article/B6VRT-4F13RF4-H/2/cde6be3ccb88627df284dfc664f68fe1>

King K, Kang H, Jin M & Lew DJ (2013) Feedback control of Swe1p degradation in the yeast morphogenesis checkpoint. *Mol. Biol. Cell* **24**: 914–22 Available at: <http://www.molbiolcell.org/content/24/7/914.short> [Accessed May 28, 2014]

Koren A, Soifer I & Barkai N (2010) MRC1-dependent scaling of the budding yeast DNA replication timing program. *Genome Res.* **20**: 781–90 Available at: <http://www.pubmedcentral.nih.gov/articlerender.fcgi?artid=2877575&tool=pmcentrez&renderType=abstract> [Accessed July 17, 2011]

Moore SA (1988) Kinetic evidence for a critical rate of protein synthesis in the *Saccharomyces cerevisiae* yeast cell cycle. *J. Biol. Chem.* **263**: 9674–9681

Moretto F, Sagot I, Daignan-Fornier B & Pinson B (2013) A pharmaco-epistasis strategy reveals a new cell size controlling pathway in yeast. *Mol. Syst. Biol.* **9**: 707 Available at: <http://www.ncbi.nlm.nih.gov/pubmed/24217298> [Accessed March 28, 2014]

Ohya Y, Sese J, Yukawa M, Sano F, Nakatani Y, Saito TL, Saka A, Fukuda T, Ishihara S, Oka S, Suzuki G, Watanabe M, Hirata A, Ohtani M, Sawai H, Fraysse N, Latge J-P, Francois JM, Aebi M, Tanaka S, et al (2005) High-dimensional and large-scale phenotyping of yeast mutants 10.1073/pnas.0509436102. *Proc. Natl. Acad. Sci. U. S. A.* **102**: 19015–19020 Available at: <http://www.pnas.org/cgi/content/abstract/102/52/19015>

Pfeuty B & Kaneko K (2007) Minimal requirements for robust cell size control in eukaryotic cells. *Phys. Biol.* **4**: 194–204 Available at: http://www.ncbi.nlm.nih.gov/entrez/query.fcgi?cmd=Retrieve&db=PubMed&dopt=Citation&list_uids=17928658

Polymenis M & Schmidt E V (1997) Coupling of cell division to cell growth by translational control of the G1 cyclin CLN3 in yeast. *Genes Dev.* **11**: 2522–2531

Popolo L, Vanoni M & Alberghina L (1982) Control of the yeast cell cycle by protein synthesis. *Exp. Cell Res.* **142**: 69–78 Available at: <http://www.sciencedirect.com/science/article/B6WFC-4DKKFG7-2V/2/9d9f6c67ac9d075ab49f961603178d95>

Rupes I (2002) Checking cell size in yeast. *Trends Genet.* **18**: 479–485 Available at: <http://www.sciencedirect.com/science/article/B6TCY-46C8PNB-1/2/941b36bd89422bf6243ef37c405e5746>

Sveiczzer A, Novák B & Mitchison JM (1996) The size control of fission yeast revisited. *J. Cell Sci.* **109**: 2947–2957

Di Talia S, Skotheim JM, Bean JM, Siggia ED & Cross FR (2007) The effects of molecular noise and size control on variability in the budding yeast cell cycle. *Nature* **448**: 947–51 Available at: <http://www.ncbi.nlm.nih.gov/pubmed/17713537> [Accessed July 25, 2011]

Truong SK, McCormick RF & Polymenis M (2013) Genetic determinants of cell size at birth and their impact on cell cycle progression in *Saccharomyces cerevisiae*. *G3 (Bethesda)*. **3**: 1525–30 Available at:

<http://www.pubmedcentral.nih.gov/articlerender.fcgi?artid=3755912&tool=pmcentrez&render type=abstract> [Accessed June 4, 2014]

Turner JJ, Ewald JC & Skotheim JM (2012) Cell size control in yeast. *Curr. Biol.* **22**: R350–9 Available at: <http://dx.doi.org/10.1016/j.cub.2012.02.041> [Accessed July 16, 2012]

Tyson JJ & Hannsgen KB (1985) Global asymptotic stability of the size distribution in probabilistic models of the cell cycle. *J. Math. Biol.* **22**: 61–68 Available at: http://www.ncbi.nlm.nih.gov/entrez/query.fcgi?cmd=Retrieve&db=PubMed&dopt=Citation&list_uids=4020305

Yu L, Pena Castillo L, Mnaimneh S, Hughes TR & Brown GW (2006) A survey of essential gene function in the yeast cell division cycle. *Mol. Biol. Cell* **17**: 4736–4747

Zhang J, Schneider C, Ottmers L, Rodriguez R, Day A, Markwardt J & Schneider BL (2002) Genomic scale mutant hunt identifies cell size homeostasis genes in *S. cerevisiae*. *Curr. Biol.* **12**: 1992–2001 Available at: <http://www.sciencedirect.com/science/article/B6VRT-47F1PMR-J/2/acd6729df87f223b80df1033401b6bb7>

2nd Editorial Decision

22 September 2014

Thank you again for submitting your work to Molecular Systems Biology. We have now heard back from the two referees who accepted to evaluate the revised study. As you will see, the referees are now overall supportive. They still raise some remaining concerns which should be addressed with suitable amendments in the text and toning down some of the conclusions.

Reviewer #1:

The authors did a good job addressing most of the previous concerns. I think this paper should be published because the datasets they have generated will be of great interest to the field and the results are presented clearly, in the text and in figures.

MINOR POINTS

1. I maintain my reservations about their conclusions regarding the rpl vs. rps differences (their conclusions were derived from about a third of such mutants, and not from the whole set). I suggest that it would be prudent to make it clearer in the text, especially in the discussion, that their conclusions were not derived from examination of all rpl and rps mutants, but from a subset.
2. As they describe in the revised version, the correlations reported in Fig. 1 were from Pearson tests, which is fine for straightforward linear correlation of two variables. However, given the huge variability of methods used and the absolute values of the variables coming out of each study they compare, it is again more prudent to also use and report an ordinal, non-parametric correlation test. This is easy to do, and it might yield some unexpected results.
3. Their conclusions about growth rate effects were really only from mutants within a limited range of growth rates (80-120 min doubling time). They state this, but it should be made clearer in several points in the text, when they report and discuss relevant data. Some could argue that this is a very narrow dynamic range, and if they went beyond that, their conclusions would be significantly different.

Reviewer #3:

The revised manuscript by Soifer and colleagues is significantly improved. The manuscript has been largely rewritten to enhance clarity and highlight the major findings and their relationship to previous literature. Many of the figures have also been revised either in terms of labeling or presentation to clarify the major conclusions. In addition, each of the major conclusions is clearly stated and then independently discussed in the Discussion. Finally, the correction of many errors in grammar and usage has significantly increased readability. As a consequence, this reviewer's understanding of the authors' interpretation of the data is greatly enhanced.

Having said that, the major conclusions are more reinterpretations than novel findings. The authors now acknowledge that their primary findings, the relationship between birth size and size at budding, the identity of most of the genes affecting that relationship and the existence of a weak cell size regulation outside of G1 phase, are not entirely novel but size control has been evaluated and interpreted in a novel manner. The distinction between the effects of mutants in large and small ribosome subunit constituents on cell size and the identification of a few new mutants affecting cell size are novel findings. Although neither of these is investigated in sufficient depth to understand the mechanism, the authors do propose one reasonable explanation for the observations concerning ribosomal proteins. Nevertheless, those findings, along with the expanded genomic screen and the increased discrimination of that screen in discovering cell size determination mutants warrants the interest of yeast cell biologists and others interested in the problem of cell size determination.

With those reservations, I can recommend the paper for publication in MSB.

2nd Revision

11 October 2014

Reviewer #1:

The authors did a good job addressing most of the previous concerns. I think this paper should be published because the datasets they have generated will be of great interest to the field and the results are presented clearly, in the text and in figures.

We thank the reviewer for insightful suggestions that improved the paper.

MINOR POINTS

1. I maintain my reservations about their conclusions regarding the rpl vs. rps differences (their conclusions were derived from about a third of such mutants, and not from the whole set). I suggest that it would be prudent to make it clearer in the text, especially in the discussion, that their conclusions were not derived from examination of all rpl and rps mutants, but from a subset.

We modified the corresponding paragraph in the discussion as follows (underlined text):

Our results suggest a unified explanation for those findings. Upon screening of approximately one-third of nonessential ribosomal proteins, we observed that deleting components of the large ribosomal subunit or of genes involved in ribosomal biogenesis does not affect the actual size control. Rather, cells become smaller simply because they grow slower. In contrast, parts of the small ribosomal subunit behave as positive regulators of START, extending G1 duration more than expected given their birth size, and consequently budding at a size comparable or larger than wild type cells born at a small size. This likely reflects their distinct role in translation initiation, not shared by the large ribosomal subunit. Our results therefore suggest that translation initiation is a positive regulator of START, hence its inhibition, as observed in the initial experiments, prolong G1 and could lead to a larger budding size. In contrast, translation elongation affects predominantly the cell growth rate, and therefore decreases cell size, as observed upon deletion of ribosomal components.

2. As they describe in the revised version, the correlations reported in Fig. 1 were from Pearson tests, which is fine for straightforward linear correlation of two variables. However, given the huge variability of methods used and the absolute values of the variables coming out of each study they compare, it is again more prudent to also use and report an ordinal, non-parametric correlation

test. This is easy to do, and it might yield some unexpected results.

We added expanded view subplot E1F that shows correlations as in Fig. 1 but using Spearman correlation coefficient instead of Pearson. The plot is cited also in the main text.

3. Their conclusions about growth rate effects were really only from mutants within a limited range of growth rates (80-120 min doubling time). They state this, but it should be made clearer in several points in the text, when they report and discuss relevant data. Some could argue that this is a very narrow dynamic range, and if they went beyond that, their conclusions would be significantly different.

We modified the relevant paragraph in the discussion as follows (additions underlined)
Examining size control in wild type cells growing at different rates suggested a way to distinguish between the direct and the indirect (e.g. acting through growth rate) effects on size regulation. When considering cells born at the same size (but growing at different rates), the increase in volume during G1 was strongly dependent on the proliferation rate. In contrast, the duration of G1 was largely defined by the initial cell size, independently of the cell growth rate, at least for growth rate interval of 86-124 which we have checked and where the vast majority of wild-type cells reside. We therefore considered the size-dependent regulation of G1 length as the primary mode by which cells guard against size fluctuations, and examined for mutants that alter this dependency.

Reviewer #3:

The revised manuscript by Soifer and colleagues is significantly improved. The manuscript has been largely rewritten to enhance clarity and highlight the major findings and their relationship to previous literature. Many of the figures have also been revised either in terms of labeling or presentation to clarify the major conclusions. In addition, each of the major conclusions is clearly stated and then independently discussed in the Discussion. Finally, the correction of many errors in grammar and usage has significantly increased readability. As a consequence, this reviewers understanding of the authors interpretation of the data is greatly enhanced.

We appreciate the reviewer's insightful comments and thorough review.